🔓 | **Open Peer Review** | Environmental Microbiology | Research Article

# Diversity and impact of single-stranded RNA viruses in Czech *Heterobasidion* populations

**László Benedek Dálya,**[1] **Martin Černý,**[2] **Marcos de la Peña,**[3] **Anna Poimala,**[4] **Eeva J. Vainio,**[4] **Jarkko Hantula,**[4] **Leticia Botella**[1]

**ABSTRACT**    *Heterobasidion annosum sensu lato* comprises some of the most devastating pathogens of conifers. Exploring virocontrol as a potential strategy to mitigate economic losses caused by these fungi holds promise for the future. In this study, we conducted a comprehensive screening for viruses in 98 *H. annosum* s.l. specimens from different regions of Czechia aiming to identify viruses inducing hypovirulence. Initial examination for dsRNA presence was followed by RNA-seq analyses using pooled RNA libraries constructed from *H. annosum* and *Heterobasidion parviporum*, with diverse bioinformatic pipelines employed for virus discovery. Our study uncovered 25 distinct ssRNA viruses, including two ourmia-like viruses, one mitovirus, one fusarivirus, one tobamo-like virus, one cogu-like virus, one bisegmented narna-like virus and one segment of another narna-like virus, and 17 ambi-like viruses, for which hairpin and hammerhead ribozymes were detected. Coinfections of up to 10 viruses were observed in six *Heterobasidion* isolates, whereas another six harbored a single virus. Seventy-three percent of the isolates analyzed by RNA-seq were virus-free. These findings show that the virome of *Heterobasidion* populations in Czechia is highly diverse and differs from that in the boreal region. We further investigated the host effects of certain identified viruses through comparisons of the mycelial growth rate and proteomic analyses and found that certain tested viruses caused growth reductions of up to 22% and significant alterations in the host proteome profile. Their intraspecific transmission rates ranged from 0% to 33%. Further studies are needed to fully understand the biocontrol potential of these viruses *in planta*.

**IMPORTANCE**    *Heterobasidion annosum sensu lato* is a major pathogen causing significant damage to conifer forests, resulting in substantial economic losses. This study is significant as it explores the potential of using viruses (virocontrol) to combat these fungal pathogens. By identifying and characterizing a diverse array of viruses in *H. annosum* populations from Czechia, the research opens new avenues for biocontrol strategies. The discovery of 25 distinct ssRNA viruses, some of which reduce fungal growth and alter proteome profiles, suggests that these viruses could be harnessed to mitigate the impact of *Heterobasidion*. Understanding the interactions between these viruses and their fungal hosts is crucial for developing effective, environmentally friendly methods to protect conifer forests and maintain ecosystem health. This study lays the groundwork for future research on the application of mycoviruses in forest disease management.

**KEYWORDS**    growth rate, *Heterobasidion annosum*, mycovirus, proteomics, root rot, ssRNA

**Ad Hoc Peer Reviewer** İkbal Agah İnce, University Medical Center Groningen, Groningen, the Netherlands

Address correspondence to Leticia Botella, qqbotell@mendelu.cz.

The authors declare no conflict of interest.

See the funding table on p. 24.

T he *Heterobasidion annosum sensu lato* species cluster comprises fungal root rot pathogens in the temperate and boreal regions of the Northern Hemisphere (1). These basidiomycetes represent a major threat to intensively managed forest stands

while playing a subordinate role in natural ecosystems (2, 3). Financial losses attributable to the pathogen in the European Union are approximately 1.4 billion € annually, inflation-adjusted from (4). Heterobasidion root rot has been a serious problem in planted Norway spruce (*Picea abies*), and Scots pine (*Pinus sylvestris*) stands in Czechia ever since forest management was introduced (5, 6) and is one of the significant factors that expose trees to European spruce bark beetle (*Ips typographus*) infestation (7). In 2020, the volume of salvage fellings in Czechia rose to an unprecedented 34 million m$^3$, mostly due to *I. typographus* as the main disease agent (8). Despite the extent of damages, no practical measures have been implemented in Czechia to control the spread of *H. annosum* s.l. because the problem generally receives little attention.

Preventive control, that is, chemical (urea, borax) or biological (antagonistic fungus *Phlebiopsis gigantea*) stump treatments against aerial spore infection should be a high priority in disease management strategies against *Heterobasidion* spp. (9). However, this is of little avail in many regions due to the nearly ubiquitous occurrence of the pathogen (10). Currently available curative control methods are rotation of tree species at a site and stump removal (1). Change of the tree species is not always possible due to site characteristics or economic considerations. Stump harvesting removes only a fraction of the pathogen inoculum (11) and may lead to several negative environmental impacts (12). The eradication of the fungus, once it has established itself in a stand, can be a very lengthy, laborious, and costly process with varying success rates. During the past 25 years, the possibility of using mycoviruses as biocontrol agents was being explored in Sweden (e.g., 13) and Finland (e.g., 14) as a fundamentally new solution to limit *Heterobasidion* damages.

Viruses that inhabit fungi, that is, mycoviruses, are hosted by a wide range of fungal taxa (15). The high phylogenetic diversity of mycoviruses is reflected by their current classification into 10 double-stranded (ds) RNA families, one unclassified dsRNA genus, 15 single-stranded (ss) RNA families, and one ssDNA family (https://ictv.global/), and many novel taxa are awaiting description. The discovery rate of new viruses is exponentially accelerated by high-throughput sequencing approaches (16). The incidence of mycoviral infections in plant-associated fungi ranges from a few % to over 90% (17). Most of these virus/host interactions are cryptic, but beneficial or harmful effects are also known. Mycoviruses that induce hypovirulence in phytopathogenic fungi may be utilized as biological control agents. The best-known example is Cryphonectria hypovirus 1 (CHV1), which was successfully applied as a treatment against chestnut blight in Europe (18). In addition, virus-induced hypovirulence has been shown in *Rosellinia necatrix*, *Sclerotinia sclerotiorum*, *Botryosphaeria dothidea,* and *Fusarium* spp., among other fungal pathogens (19).

The *Heterobasidion* genus hosts a diverse mycovirus community with worldwide distribution. Viruses with dsRNA genomes infect approximately 15% of *H. annosum* s.l. isolates in Europe and Western Asia (20, 21). The most common mycovirus in *Heterobasidion* spp., Heterobasidion RNA virus 6 (HetRV6; *Curvulaviridae* family), is responsible for more than 70% of all dsRNA infections (22). HetRV6 was found in many different *Heterobasidion* species of distant origins (e.g., Europe, Siberia, and USA) and showed a high degree of geographic and host-related differentiation (22). Members of *Partitiviridae* are also widespread and highly diverse. So far, 21 different partitiviruses have been described in *Heterobasidion* spp., and all belong to the *Alphapartitivirus* or *Betapartitivirus* genera (23). Mitoviruses (*Mitoviridae* family) were detected by small RNA deep sequencing and RNA-seq in *Heterobasidion parviporum* and *H. annosum* (24, 25). More recently, further positive-strand RNA viruses, representing the *Narnaviridae* and *Botourmiaviridae* families were discovered in *H. parviporum*, as well as members of the newly established *Ambiviricota* phylum (*Dumbiviridae* and *Trimbiviridae* families) with ambisense RNA genomes (23, 26).

Ample evidence shows that virus transmission is possible within and between *Heterobasidion* species both *in vitro* (14, 21, 22, 27–31) and in natural conditions (23, 32–34). Pre-existing viruses in either the donor or the recipient mycelium may help or hinder

the transmission of other viruses (30, 31, 33). Although most viruses of *Heterobasidion* are considered more or less latent, both negative and slightly positive interactions with the fungal hosts have been reported (14, 22, 28, 30, 35–37). The same mycovirus can exhibit differential effects on its hosts depending on the particular host strain and environmental conditions (28, 36). Laboratory and field experiments with Heterobasidion partitivirus 13 from *H. annosum* (HetPV13-an1) turned out promising for the development of a future biocontrol product (14, 30, 31). By affecting the transcription of 683 genes (representing 6% of the host genome), the virus influenced the cell cycle, hampered the uptake of energy, caused metabolic changes, limited the growth of host isolates, and altered mycelial morphology (14). As a result, the virulence of its natural host, as well as that of the recipient *H. parviporum* strains reduced considerably. Very recently, HetPV13-an1 was shown to enhance the biocontrol effect of *P. gigantea* when applied to pine stumps infested by *H. annosum* (34).

As previous research focused on boreal forests, limited data are available about viral communities hosted by Central European *Heterobasidion* strains. However, of dsRNA viruses, HetRV6 is known to occur in Austria and Poland (*Heterobasidion abietinum, H. parviporum,* and *H. annosum*), and the partitiviruses HetPV11 (several isolates), HetPV12-an1, as well as the putative biocontrol virus HetPV13-an1 originate from Poland (*H. annosum*). We hypothesized that Czech *Heterobasidion* isolates harbor similar viruses. The specific aims of this study were to: (i) collect ca. 100 strains of *H. annosum* s.l. from different regions of Czechia, (ii) screen the gathered isolates for virus presence by both traditional and state-of-the-art methods, (iii) describe the putative mycoviruses and characterize their genome, and (iv) investigate the effects of selected viruses on the phenotype of their host. Depending on the outcomes of these experiments, the long-term objective is to explore the possibilities of virocontrol against Heterobasidion root rot with virus strains naturally distributed in Czechia. The present work might represent an early step towards restoring the health of the remaining conifer stands infested by *Heterobasidion*.

## MATERIALS AND METHODS

### Origin of the fungal isolates

Strains of *H. annosum* s.l. were collected by arbitrary sampling throughout Czechia, with an emphasis on managed stands of Norway spruce and Scots pine. Fruiting bodies were gathered from substrates (decayed stumps, logs, roots, or the ground) located at least 30 m apart to minimize the chance of repeated sampling of the same genotype (38). Pure cultures were isolated from basidiocarps and grown on 2% malt extract agar (MEA; HiMedia, India) at 18°C. One genotype was obtained from a wood disc used as a spore trap, by isolating an emerging conidiophore. In total, 96 *Heterobasidion* strains were collected from Czechia, and two Slovakian strains were included in the study as well (Table S1 available at https://doi.org/10.6084/m9.figshare.26503525).

Isolates were identified at the species level as described earlier (10). For isolates collected in 2018–2019, the dilution protocol of the Phire Plant Direct PCR Kit (Thermo Fisher Scientific, Waltham, MA, USA) was followed with 2 µL supernatant as a template. Mycelia were subcultured onto MEA covered by a thin cellophane membrane, incubated at 22°C in the dark for 3 weeks, harvested into sterile 50 mL Falcon tubes, lyophilized at −55°C for 20 h, and pulverized with two autoclaved 10 mm steel balls by vortexing for 2 min.

### RNA extractions

The presence of dsRNA elements in the isolates was examined using CF11 cellulose affinity chromatography (39), modified by (40), using 2 g of fresh mycelium. The size of the dsRNAs was estimated by electrophoresis using a 1.2% agarose gel. Total RNA was extracted by the Quick-RNA Fungal/Bacterial Miniprep Kit (Zymo Research, Irvine, CA,

USA) from the fungal cultures grown and prepared as described in the previous section. Homogenization was done by shaking for 2 min at 30 Hz in a Mixer Mill MM 400 (Retsch, Haan, Germany). The quality of total RNA was visually assessed using a 1.2% agarose gel. RNA concentrations were measured by a Qubit 2.0 Fluorometer (Invitrogen). The RNA solutions were stored at −80℃.

## RNA sequencing

Three pools consisting of total RNA of 16 *H. annosum* strains, another 16 *H. annosum* strains, and 13 *H. parviporum* strains (Table S1 available at https://doi.org/10.6084/m9.figshare.26503525) were prepared as follows. Equal volumes of each RNA sample were mixed, diluted to 200 ng/µL in 70 µL volume, treated with the TURBO DNA-free Kit (Thermo Fisher Scientific), and sent to SEQme s.r.o. (Dobříš, Czechia). Ribosomal RNA was removed using the NEBNext rRNA Depletion Kit (Human/Mouse/Rat). The cDNA libraries were constructed using the NEBNext Ultra II Directional RNA Library Prep Kit for Illumina and sequenced in paired-end (2 × 150 bp) on a NovaSeq 6000 (Illumina). After the confirmation of multiple virus infections in *H. annosum* isolate 1987, RNA-seq was repeated for this single strain as described above, with the following exceptions. Total RNA was isolated using RNAzol RT (Sigma-Aldrich, Steinheim, Germany) with homogenization lasting 1 min. DNase treatment was omitted, and RNA-seq was performed by the Institute of Applied Biotechnologies a.s. (Prague, Czechia) with a read length of 2 × 151 bp.

## Bioinformatics

Two approaches were combined for *in silico* data mining for viruses in all of the NGS data sets, each of which consisted of ca. 600 million reads. The quality of raw reads was assessed using FastQC-0.11.8 (https://www.bioinformatics.babraham.ac.uk/projects/fastqc, accessed on 11 April 2022). Adapter sequences were clipped with cutadapt-3.4 (41), and zero-length reads were discarded. The first bioinformatic pipeline consisted of *de novo* assembly with Trinity-2.11.0 (42) using RF for—SS_lib_type flag and searching for similarities of the obtained contigs to custom virus and host protein and nucleotide reference databases obtained from NCBI, using the BLASTX and BLASTN algorithms of the BLAST +−2.10.0 program (43). The E value threshold was 1e-5. Contigs showing homologies to viral proteins or genomic sequences were selected for further analysis, whereas contigs with homologies to the host genome were excluded.

In the second pipeline, reads mapped to the host genome with BWA-0.7.12-r1034 MEM algorithm (44) were removed using SAMtools-1.9 (45). The BamTools-2.5.1 suite (46) was used for generating mapping statistics and for file format conversion. Mapping quality was also evaluated by QualiMap-2.2.2, BAM QC mode (47). The host-unmapped reads were aligned to a nucleotide database of *Heterobasidion* viruses with BWA MEM. Alignments were visualized in Tablet-1.12.02.08 (48). Random sequencing errors in the host-unmapped reads were fixed with Rcorrector-1.0.4 (49). Read pairs where at least one read was deemed unfixable were discarded using the "FilterUncorrectabledPE-fastq.py" python script (https://github.com/harvardinformatics/TranscriptomeAssembly-Tools/blob/master/FilterUncorrectabledPEfastq.py, accessed on 11 April 2022). The *de novo* assembly of host-unmapped reads and the subsequent similarity searches were conducted as in the first pipeline.

The lists of virus candidate contigs identified via both pipelines, together with their best hits in BLASTX and BLASTN, were examined further. Final viral contigs were selected based on sequence similarity to known viruses, identification of conserved viral marker genes (e.g., capsid protein, RdRp), and coverage depth across the contig to ensure it represents a complete or near-complete viral genome. Within each of these groups, one contig deemed the most representative of the putative mycovirus according to its length and alignment parameters was imported to Geneious Prime 2021.1.1 (https://www.gene-ious.com). All contigs from both Trinity assemblies of all three pooled data sets were mapped against each selected contig using Geneious mapper with medium sensitivity.

After the manual correction of misalignments, the resulting consensus sequences were saved and used in further analyses.

The screening of small self-cleaving ribozymes was carried out using INFERNAL software (50), which identifies RNA elements using secondary structure and nucleotide covariation modeling. Covariance models for the small ribozymes recently described from diverse ambiviruses, mitoviruses, and other viroid-like agents were used for the searches (see 51).

## RT-PCR screening

cDNA was synthesized from the total RNA samples using either the ProtoScript II First Strand cDNA Synthesis Kit (New England Biolabs, Ipswich, MA, USA), following the standard protocol with Random Primer Mix or the High-Capacity cDNA Reverse Transcription Kit (Thermo Fisher Scientific). To reveal the identity of virus-hosting *Heterobasidion* strains, RT-PCRs of the strains belonging to the RNA pool in which the putative virus was detected (Table S1 available at https://doi.org/10.6084/m9.figshare.26503525) were performed with virus-specific consensus primers (Table S2 available at https://doi.org/10.6084/m9.figshare.26503525). These primers were designed by Primer3-2.3.7 under Geneious to amplify a fragment within an ORF of putative viruses. Amplifications were carried out by the Phire Green Hot Start II PCR Master Mix (Thermo Fisher Scientific) or the OneTaq Quick-Load 2X Master Mix with Standard Buffer (New England Biolabs) with 2.5 µL cDNA in 25 µL reaction volume. PCR products were electrophoresed in a 1% agarose gel (Fig. S1). When multiple bands were visible, the targeted one was purified from a gel using the NucleoSpin Gel and PCR Clean-up (MACHEREY-NAGEL, Düren, Germany) or the Monarch DNA Gel Extraction Kit (New England Biolabs). In a few cases, a second PCR with 0.5–1 µL of the PCR product as a template was necessary to obtain a band with sufficient concentration. Target amplicons were sequenced by Eurofins Genomics (Ebersberg, Germany).

DNA was extracted from lyophilized mycelium of isolate 1987 with the DNeasy Plant Mini Kit (QIAGEN, Valencia, CA, USA) and used as a template in a PCR to rule out the integration of one movement protein (MP) encoding viral segment in the host genome (see section 3.8).

Additionally, 85 *H. annosum* s.l. isolates (Table S1 available at https://doi.org/10.6084/m9.figshare.26503525) were screened for the presence of HetRV6 with specific primers designed by (22). Here, 2 µL cDNA was used as a template in RT-PCR; the rest of the workflow was as described above.

## Confirmation of the 3′ end of an incomplete viral genome

Rapid amplification of cDNA ends (RACE) of the incomplete RNA-dependent RNA polymerase (RdRp) encoding region at the 3′ terminus of a putative linearized ambi-like virus was done following the protocol of the SMARTer RACE 5′/3′ Kit (Takara Bio USA, Mountain View, CA, USA), employing a specific primer designed by Primer3-2.3.7 under Geneious (Table S2 available at https://doi.org/10.6084/m9.figshare.26503525). PolyA RNA was purified with the RNA Clean & Concentrator-5 Kit (Zymo Research) and quantified using a BioSpec-nano spectrophotometer (Shimadzu, Japan). The RACE product was cloned using the In-Fusion HD Cloning Kit (Takara Bio USA) following the producer's instructions and modifications as described by (52).

## Genetic variability and phylogenetic analyses

For calculating the genetic distance between mycovirus strains, pairwise (pw) sequence alignments of Sanger sequenced amplicons, as well as their translations, were done by MAFFT-7.490 (53) under Geneious. Maximum likelihood phylogenetic trees based on the amino acid (aa) sequences of either the RdRp or the MP were constructed using the bootstrapping algorithm implemented in MEGA11 (54). The tree for the *Virgaviridae* family was built based on the codon-aligned nucleotide sequences of the replication

proteins and RdRps. The model with the lowest Bayesian Information Criterion score was selected for each tree.

## Virus elimination

To assess the effect of mycoviruses on their host, it was necessary to create isogenic lines of selected fungal isolates with and without virus(es). In an attempt to eliminate viruses, 10 monohyphal cultures of three naturally infected *H. annosum* isolates (1987, 1989, and 2072) were generated as follows. Hyphal tips of isolates grown on 3% water agar were picked with a Pasteur pipette under a microscope and inoculated onto MEA plates. Subcultures of the resulting single hyphal isolates were cultivated on cellophane-covered MEA plates for ca. 2 weeks. Total RNA was extracted from the harvested mycelia by the E.Z.N.A. Fungal RNA Mini Kit (Omega Bio-tek, Norcross, GA, USA), following the standard protocol. The fungal tissue was disrupted by a FastPrep-24 Classic homogenizer (MP Biomedicals, Santa Ana, CA, USA) and 1–2 mm quartz sand grains. The RNA concentration and integrity were measured with a NanoDrop One$^C$ spectrophotometer (Thermo Fisher Scientific). cDNA was synthesized from 3 µg RNA using RevertAid Reverse Transcriptase (Thermo Fisher Scientific) and random hexamer primers with a denaturation step at 98°C for 5 min. The presence of viruses in the single hyphal cultures was confirmed via RT-PCR, using DreamTaq Green DNA Polymerase (Thermo Fisher Scientific) with 1 µL cDNA in 20 µL reaction volume. PCR products were electrophoresed in a 1% agarose gel. When multiple bands appeared, the targeted one was purified from a gel using the E.Z.N.A. Gel Extraction Kit (Omega Bio-tek). Target amplicons were sequenced at Macrogen Europe (Amsterdam, The Netherlands).

As the above method was only partially successful, thermal treatment was applied for virus removal (14, 28). Four monohyphal cultures were derived from isolate 1987, each having retained different combinations of viruses, and the original isolates 1987, 1989, and 2072 were cultivated on MEA at 32°C for 1 week and at 33°C for another week. Subcultures were taken at the end of both weeks, allowed to recover to a normal growth rate at 20°C, and subjected to RT-PCR screening.

## Horizontal transmission

Virus transmission was expected to occur via hyphal anastomosis in dual cultures. Each of three virus-infected (1987, 1989, and 2072) *H. annosum* isolates was paired with three virus-free (1985, 2013, and 2052) *H. annosum* strains on 9 cm MEA plates and incubated at 22°C in the dark for 1 month. Small agar pieces taken from the recipient side were transferred to new MEA plates partially covered by cellophane. The sampling spots were located (i) within 1 cm from the interaction zone separating the donor and the recipient strain and (ii) at ca. 2 cm from the interaction zone (Fig. S2A). After 2–5 weeks, fresh mycelium was ground in liquid N$_2$ using sterile mortar and pestle, and total RNA was extracted using RNAzol RT. cDNA synthesis and RT-PCR were conducted for the initial screening. The whole experiment was repeated using a replicate plate of the same pairings incubated for 3 months. This time, the following spot was also sampled: (iii) far from the interaction zone, near the edge of the Petri dish (Fig. S2B).

The genetic identity of the virus-infected recipient isolates was verified with random amplified microsatellite (RAMS) fingerprinting with CCA primer (55). DNA was isolated from fresh mycelium using the DNeasy Plant Mini Kit with 2 × 50 µL elution volume. Amplifications were performed in two repetitions by the OneTaq Quick-Load 2X Master Mix with standard buffer, with 2 µM primer and 0.5 µL DNA in 25 µL reaction volume. Thermal cycling was conducted as follows: 4 min at 94°C, followed by 35 cycles of 30 s at 94°C, 45 s at 55°C, and 2 min at 68°C, with a final extension of 5 min at 68°C. PCR products were electrophoresed in a 1.8% agarose gel.

## Growth rate measurements

Fifteen *H. annosum* isolates were selected for growth rate comparison *in vitro*. Circular agar plugs with fresh mycelium were inoculated onto the center of 9 cm MEA plates

(five replicates) and incubated at 22°C in the dark. The mycelial edge was marked at the bottom of the plates 3 and 5 days post-inoculation (dpi) and subsequently photographed. The area of fungal colonies was measured using ImageJ-1.53v (56). Comparison of the absolute growth of *a priori* determined pairs of isolates at 3 and 5 dpi was done by Student's *t*-test. The levels of significance were defined as $P < 0.05$ and $P < 0.001$. Statistical analyses and data visualization were performed in R Studio (57)(R Core Team, 2022; https://www.r-project.org).

## Proteomic analysis

The cellular response of *H. annosum* to mycoviral infections was investigated by proteomics. Cellophane-covered MEA plates (four replicates) were inoculated with the same fungal strains as in the growth rate experiment. After incubation at 22°C in the dark for 12 days, mycelia were harvested into sterile 15 mL Falcon tubes and lyophilized at −55°C for 4 h. Total protein extracts were prepared as previously described (e.g., 58), and portions of samples corresponding to 5 µg of peptide were analyzed by nanoflow reverse-phase liquid chromatography-mass spectrometry using a 15 cm C18 Zorbax column (Agilent, Santa Clara, CA, USA), a Dionex Ultimate 3000 RSLC nano-UPLC system, and the Orbitrap Fusion Lumos Tribrid Mass Spectrometer equipped with a FAIMS Pro Interface. All samples were analyzed using FAIMS compensation voltages of −40,−50, and −75 V. The measured spectra were recalibrated, filtered (precursor mass: 350–5,000 Da; S/N threshold: 1.5), and searched against the *Heterobasidion annosum* v2.0 (59) and common contaminants databases using Proteome Discoverer 2.5 (Thermo Fisher Scientific, algorithms SEQUEST; MS Amanda, 60). The quantitative differences were determined by Minora, employing precursor ion quantification followed by normalization (total area) and calculation of relative peptide/protein abundances. The reported statistical tests were generated and implemented as follows using the default and recommended settings. The reliability of the protein identifications was assessed in Proteome Discoverer 2.5. The Student's *t*-test was calculated using MS Excel. For the ANOVA with Tukey's Tukey's Honestly Significant Difference (HSD) and the Kruskal-Wallis tests, the Real Statistics Resource Pack software for MS Excel (Release 6.8; Copyright 2013–2020; Charles Zaiontz; https://www.real-statistics.com) and MetaboAnalyst 5.0 (61) were employed. Orthogonal Partial Least Squares (OPLS0) and Variable Importance in Projection (VIP) were performed in SIMCA 14.1 (Sartorius, Goettingen, Germany). Significant differences refer to $P < 0.05$.

## RESULTS AND DISCUSSION

### Virus detection and discovery

The applied methodology allowed for the reliable detection of known viruses, as well as the discovery of novel viruses. No dsRNA elements were found in our collection of 98 *Heterobasidion* isolates analyzed using cellulose chromatography. In agreement with this result, HetRV6 was absent in a subset of 85 isolates screened with RT-PCR. Altogether, 25 ssRNA viruses were discovered in the RNA-seq data sets comprising altogether 32 *H. annosum* and 13 *H. parviporum* isolates (Table 1). Twenty-seven percent of these isolates were infected by at least one virus. All detected viruses were hosted by *H. annosum*, except one strain of the recently described Heterobasidion ambi-like virus 3 (HetAlV3) (23), which dwelt in *H. parviporum* (Table 1). The presence of each virus was verified in up to five host strains. Six isolates harbored a single virus, whereas coinfections were recorded in another six isolates. The most salient example was 1987 (Table 1), a strain hosting 10 viruses belonging to four taxonomic groups. The highest documented natural virus load in *Heterobasidion* consisted of eight dsRNA or ssRNA viruses in a single *H. parviporum* strain (23), whereas in a transmission experiment, the highest achieved number of coinfecting dsRNA viruses was five (31). Infections with comparably high numbers of related and unrelated viruses are known in other fungi (62, 63) and oomycetes (64, 65).

**TABLE 1** Mycoviruses detected

| Mycovirus | Acronym | Segment | GenBank accession | Host strain | Length (nt) | % GC | Mapping reads[h,i] | Mean depth[i] | BLASTX first hit by total score[i] | Identity | Query cover | E value[j] |
|---|---|---|---|---|---|---|---|---|---|---|---|---|
| Heterobasidion mitovirus 3 | HetMV3 | RNA1 | ON014534 | H. annosum 2009 | 4,933 | 43.1% | 228,580 | 6,850 | Heterobasidion mitovirus 3 (QED55407)[k] | 96.9% | 49%[k] | 0[k] |
| Heterobasidion narna-like virus 2 | HetNlV2 | RNA1 | ON014535 | H. annosum 1987[a] | 3,877 | 53.7% | 57,684/51,735 | 2,187/1,919 | Heterobasidion narna-like virus 1 (UHK02569) | 80.3% | 94% | 0 |
| Heterobasidion narna-like virus 4 | HetNlV4 | RNA1 | ON985234 | H. annosum 1987[a] | 3,970 | 54.0% | 62,816/42,325 | 2,329/1,538 | Heterobasidion narna-like virus 1 (UHK02569) | 80.2% | 95% | 0 |
| | | RNA2 | ON014536 | | 4,011 | 55.5% | 40,456/150,103 | 1,484/5,410 | Heterobasidion narna-like virus 1 (UHK02570) | 60.5% | 87% | 0 |
| Heterobasidion ourmia-like virus 2 | HetOlV2 | RNA1 | ON014537 | H. annosum 2072 | 2,783 | 57.4% | 438 | 23 | Leucocoprinus ourmiavirus B (QED42953) | 51.8% | 46% | 8e−111 |
| Heterobasidion ourmia-like virus 3 | HetOlV3 | RNA1 | ON014538 | H. annosum 2038[b] | 2,632 | 45.1% | 5,122 | 286 | Heterobasidion ourmia-like virus 1 (UHK02571) | 48.6% | 52% | 7e−110 |
| Heterobasidion fusarivirus 1 | HetFV1 | RNA1 | ON014533 | H. annosum 1987 | 6,913 | 54.4% | 444/2,101 | 9/43 | Heterobasidion parviporum fusarivirus 1 (WOK44148) | 91.5% | 82% | 0 |
| Heterobasidion tobamo-like virus 1 | HetTlV1 | RNA1 | ON014539 | H. annosum 1989 | 12,586 | 46.9% | 1,785 | 20 | Lentinula edodes tobamo-like virus 1 (QOX06054) | 35.9% | 29% | 1e−66 |
| Heterobasidion ambi-like virus 3 | HetAlV3 | RNA1 | ON014526 | H. parviporum 1991 | 4,947 | 48.7% | 740 | 22 | Heterobasidion ambi-like virus 3 (UHK02576) | 95.8% | 42% | 0 |
| Heterobasidion ambi-like virus 5 | HetAlV5 | RNA1 | ON014527 | H. annosum 2050 | 4,815 | 47.7% | 5,461 | 166 | Heterobasidion ambi-like virus 1 (UHK02572) | 76.9% | 37% | 0 |
| Heterobasidion ambi-like virus 6 | HetAlV6 | RNA1 | ON014528 | H. annosum 2028[c] | 5,108 | 46.8% | 9,624 | 276 | Armillaria novae-zelandiae ambi-like virus 1 (DAD54841) | 32.9% | 40% | 4e−94 |
| Heterobasidion ambi-like virus 7 | HetAlV7 | RNA1 | ON014529 | H. annosum 2040[d] | 4,965 | 47.1% | 8,050 | 236 | Heterobasidion ambi-like virus 4 (UHK02578) | 57.2% | 41% | 0 |

*(Continued on next page)*

**TABLE 1** Mycoviruses detected (*Continued*)

| Mycovirus | Acronym | Segment | GenBank accession | Host strain | Length (nt) | % GC | Mapping reads[h,i] | Mean depth[i] | BLASTX first hit by total score[j] | Identity | Query cover | E value |
|---|---|---|---|---|---|---|---|---|---|---|---|---|
| Heterobasidion ambi-like virus 8 | HetAlV8 | RNA1 | ON014530 | *H. annosum* 1983[e] | 5,026 | 47.5% | 8,346 | 243 | Heterobasidion ambi-like virus 4 (UHK02578) | 33.8% | 39% | 2e−90 |
| Heterobasidion ambi-like virus 9 | HetAlV9 | RNA1 | ON014531 | *H. annosum* 2028 | 4,903 | 48.5% | 1,140 | 34 | Heterobasidion ambi-like virus 3 (UHK02576) | 64.2% | 42% | 0 |
| Heterobasidion ambi-like virus 19 | HetAlV19 | RNA1 | ON985229 | *H. annosum* 1987 | 4,977 | 47.3% | 8,922/18,873 | 260/540 | Heterobasidion ambi-like virus 4 (UHK02578) | 56.5% | 41% | 0 |
| Heterobasidion ambi-like virus 20 | HetAlV20 | RNA1 | ON985230 | *H. annosum* 1987 | 4,919 | 48.9% | 7,517/47,340 | 225/1,389 | Heterobasidion ambi-like virus 3 (UHK02578) | 63.9% | 40% | 0 |
| Heterobasidion ambi-like virus 21 | HetAlV21 | RNA1 | ON985231 | *H. annosum* 1987 | 4,862 | 48.0% | 199,680/1,487,296 | 6,006/44,221 | Heterobasidion ambi-like virus 1 (UHK02577) | 78.0% | 37% | 0 |
| Heterobasidion ambi-like virus 22 | HetAlV22 | RNA1 | ON985232 | *H. annosum* 1987 | 4,939 | 47.4% | 26,585/185,342 | 784/5,381 | Heterobasidion ambi-like virus 4 (UHK02572) | 60.2% | 41% | 0 |
| Heterobasidion ambi-like virus 23 | HetAlV23 | RNA1 | ON985233 | *H. annosum* 1987 | 4,882 | 48.3% | 397/454 | 12/13 | Heterobasidion ambi-like virus 4 (UHK02578) | 47.4% | 41% | 0 |
| Heterobasidion ambi-like virus 24 | HetAlV24 | RNA1 | OR031238 | *H. annosum* 1993 | 4,995 | 46.9% | 8,227 | 237 | Heterobasidion ambi-like virus 4 (UHK02578) | 61.8% | 41% | 0 |
| Heterobasidion ambi-like virus 25 | HetAlV25 | RNA1 | OR031239 | *H. annosum* 1987 | 5,027 | 46.9% | 6,483/19,649 | 191/561 | Heterobasidion ambi-like virus 4 (UHK02578) | 56.7% | 41% | 0 |
| Heterobasidion ambi-like virus 26 | HetAlV26 | RNA1 | OR031240 | *H. annosum* 1983 | 5,005 | 46.4% | 14,219 | 411 | Heterobasidion ambi-like virus 4 (UHK02578) | 54.3% | 41% | 0 |
| Heterobasidion ambi-like virus 27 | HetAlV27 | RNA1 | OR031241 | *H. annosum* 1983 | 4,914 | 49.0% | 6,144 | 184 | Heterobasidion ambi-like virus 3 (UHK02577) | 61.7% | 40% | 0 |
| Heterobasidion ambi-like virus 28 | HetAlV28 | RNA1 | OR031242 | *H. annosum* 1993 | 2,537[f] | 47.4% | 4,143 | 219 | Heterobasidion ambi-like virus 4 (UHK02579) | 46.0% | 77% | 0 |

*(Continued on next page)*

**TABLE 1** Mycoviruses detected (*Continued*)

| Mycovirus | Acronym | Segment | GenBank accession | Host strain | Length (nt) | % GC | Mapping reads[h,j] | Mean depth[j] | BLASTX first hit by total score[j] | Identity | Query cover | E value |
|---|---|---|---|---|---|---|---|---|---|---|---|---|
| Heterobasidion ambi-like virus 29 | HetAlV29 | RNA1 | OR031243 | *H. annosum* 2028 | 2,390[f] | 45.9% | 1,699 | 102 | Heterobasidion ambi-like virus 4 (UHK02578) | 61.8% | 42% | 0 |
| Coguvirus movement protein-like sequence | CVMPIS | RNA2 | ON014532 | *H. annosum* 1987 | 2,425[g] | 55.5%[g] | 578 [g]/ 2,666[g] | 35 [g]/ 156[g] | Jiangsu sediment phenui-like virus (QYF49545) | 24.9% | 24% | 4e−04 |

[a]Other *H. annosum* hosts were 1983 and 1993.
[b]Other *H. annosum* hosts were 1887 and 2028.
[c]Other *H. annosum* hosts were 1983, 1993, 2038, and 2050.
[d]Another *H. annosum* host was 2028.
[e]Another *H. annosum* host was 1993.
[f]Partial genomic sequence.
[g]Without polyT tail.
[h]Using Geneious assembler with medium-low sensitivity.
[i]A single number pertains to the pooled library in which the virus was first detected and the host strain belongs; if there are two numbers, the first one pertains to library ANNO1 and the second one to library 1987.
[j]Search done on 06.01.2024.
[k]Using the Mold Protozoan Mitochondrial genetic code.

## Mitovirus

A new strain of Heterobasidion mitovirus 3 (HetMV3) was detected, sharing 97% BLASTX identity with the strain HetMV3-an1 (25; Table 1). Both strains inhabited *H. annosum*, but the virus was also recorded in *H. parviporum* (23, 25). The nucleotide (nt) sequence identities shared by the Czech HetMV3 strain and other strains originating from Poland, Finland, and Russia (25) were 94%, 90%, and 90%, respectively. This indicates intraspecies genomic differentiation with increasing geographic distance, already noted by (25). The HetMV3 genome contains the mitoviral RdRp conserved domain (Fig. 1). Phylogenetically, it clustered together with HetMV1 within the *Duamitovirus* genus of the *Mitoviridae* family (Fig. 2), confirming the analysis of (25). Despite infecting a single host strain, HetMV3 had the highest read coverage among all viruses found in our study, with a mean depth of 6850 (Table 1). Outstanding accumulation levels of HetMV3 strains relative to other viruses detected in the same RNA-Seq library have previously been observed also in *H. parviporum* (23). The high accumulation levels of Rhizoctonia cerealis duamitoviruses in their host compared with other viruses have been explained by the protective environment of mitochondria against cytoplasmic RNA interference (RNAi) (63), which was also demonstrated for Cryphonectria parasitica mitovirus 1 (66). Our data support the notion that the mitochondrial replication of mitoviruses is a successful strategy to avoid RNA silencing, although the RNAi machinery of *Heterobasidion* was shown to affect HetMV1 to a certain degree (24).

## Narna-like viruses

Two contigs resembling the RNA1 segment of Heterobasidion narna-like virus 1 (HetNlV1) (23) were found, each with 80% identity in BLASTX (Table 1). Another contig shared 61% aa identity with the hypothetical protein (HP) encoded by the HetNlV1 RNA2 segment. Similarly to HetNlV1, the length of each detected virus segment was ca. 4 kilobases (kb). All three segments had a high read coverage in data sets ANNO1 and 1987 (Table 1), and their presence was confirmed in the same three isolates. The hyphal tip isolation experiment clarified that two of the segments belong to the same virus, being always cotransmitted into the same monohyphal cultures of 1987 with a transmission frequency of 90%, as opposed to the third segment's rate of 20%. Based on these results, the unmatched RNA1 segment was designated as HetNlV2, whereas a new bisegmented narna-like virus was named HetNlV4 (Table 1; Fig. 1). The HetNlV4 RNA1 segment would have remained undetected based on analyzing only the ANNO1 library. The two

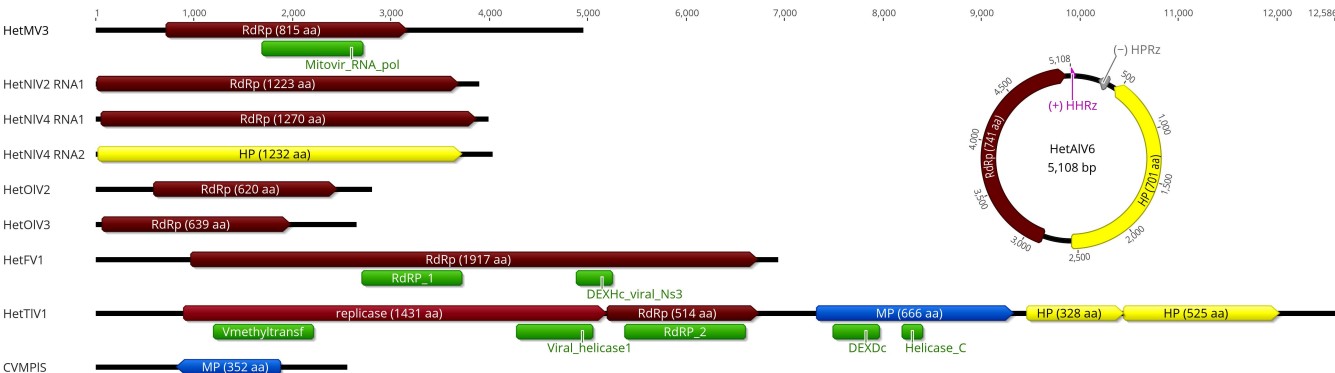

**FIG 1** Genome organization of mycoviruses detected in *Heterobasidion*, drawn to scale (indicating genome length in nucleotides). One representative of the novel ambi-like viruses was selected to illustrate the typical location of ribozymes within their circular genome. HetMV, Heterobasidion mitovirus; HetNlV, Heterobasidion narna-like virus; HetOlV, Heterobasidion ourmia-like virus; HetFV, Heterobasidion fusarivirus; HetTlV, Heterobasidion tobamo-like virus; CVMPlS, Coguvirus movement protein-like sequence; HetAlV, Heterobasidion ambi-like virus. HPRz, hairpin ribozyme; and HHRz, hammerhead ribozyme. Predicted open reading frames: RdRp, RNA-dependent RNA polymerase; HP, hypothetical protein; MP, movement protein. Conserved domains: Mitovir_RNA_pol, pfam05919 (E value: 9.17e−84); RdRP_1, pfam00680 (1.23e−15); DEXHc_viral_Ns3, cd17931 (5.37e−08); Vmethyltransf, pfam01660 (8.56e−14); Viral_helicase1, pfam01443 (4.00e−29); RdRP_2, pfam00978 (4.65e−78); DEXDc, smart00487 (2.83e−06); and Helicase_C, pfam00271 (5.78e−05).

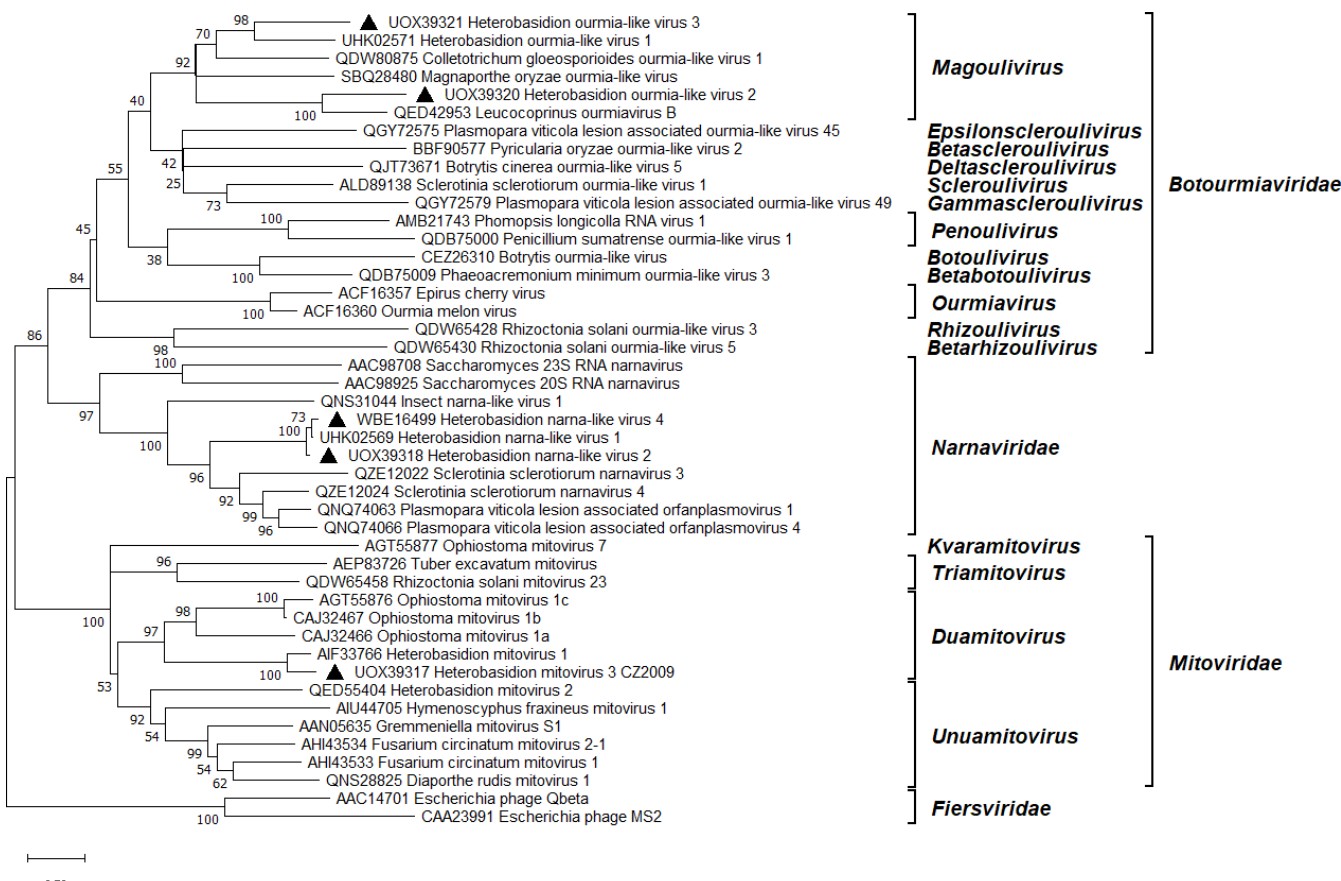

**FIG 2** Phylogenetic relationships within and between relevant families of the *Lenarviricota* phylum. The *Fiersviridae* family served as outgroup. The tree was built based on the alignment of RdRp aa sequences generated with MUSCLE in MEGA11 (54). The evolutionary history was inferred by using the maximum likelihood method and Whelan and Goldman + Freq. model (67). A discrete Gamma distribution was used to model evolutionary rate differences among sites (five categories + *G + I*). All positions with less than 95% site coverage were eliminated, that is, fewer than 5% alignment gaps, missing data, and ambiguous bases were allowed at any position. Evolutionary analyses were conducted in MEGA11 with 500 bootstrap repeats. The percentage of trees in which the associated taxa clustered together is shown next to the branches. Branch lengths are proportional to the number of substitutions per site. GenBank accessions are displayed before virus names. Ourmia- and narna-like viruses and the mitovirus strain described in this study are denoted by a triangle.

viruses shared 75% aa identity over their RdRp. Pw identities of RT-PCR amplicons of narna-like virus strains ranged 94%–99% at both the nt and aa levels (Fig. S3 available at https://doi.org/10.6084/m9.figshare.26503525). *Heterobasidion* narna-like viruses formed a highly supported cluster within *Narnaviridae* (Fig. 2). Their closest relatives were the multisegmented narna-like viruses of *S. sclerotiorum* (68) and the bipartite Plasmopara viticola lesion associated orfanplasmoviruses (69). HetNlV2 and HetNlV4 always occurred in coinfection with at least four other viruses, most frequently ambi-like viruses (Table 1), which was also the case for HetNlV1 infecting *H. parviporum* (23).

## Ourmia-like viruses

Two distinct ourmia-like viruses were discovered. For one of them, the most significant BLASTX hit was Leucocoprinus ourmiavirus B with 52% identity, whereas the other one showed the highest similarity (49% BLASTX identity) to Heterobasidion ourmia-like virus 1 (HetOlV1) (23). We named them HetOlV2 and HetOlV3, respectively (Table 1). Their nonsegmented genomes of 2.8 and 2.6 kb contain a single predicted ORF, encoding for a putative RdRp (Fig. 1). HetOlV2 dwelled in a single isolate and had a very low coverage, a mean depth of 23 reads. HetOlV3 was confirmed to infect three host isolates and reached moderate accumulation levels in the ANNO2 data set (Table 1). Pw identities of RT-PCR

amplicons of HetOlV3 strains ranged 89%–91% at the nt level and 91%–95% at the aa level (Fig. S3 available at https://doi.org/10.6084/m9.figshare.26503525). Phylogenetic analysis of the *Lenarviricota* phylum showed that *Heterobasidion* ourmia-like viruses resemble members of the *Magoulivirus* genus within *Botourmiaviridae* (Fig. 2). HetOlV3 was responsible for single or multiple infections with up to four ambi-like viruses (Table 1). In comparison, HetOlV1 in *H. parviporum* had a propensity for coinfections with various dsRNA and ssRNA viruses (23).

## Fusarivirus

We report the detection of a fusarivirus as a novelty in the virome of *Heterobasidion* spp. The relevant contig assembled from the ANNO1 data set was 3.3 kb long, representing only a partial genome lacking both ends of an ORF. Data from 1987 yielded the full genome of Heterobasidion fusarivirus 1 (HetFV1), which had the lowest read coverage among the viruses detected in this study (Table 1). HetFV1 gave highly significant matches with members of the *Fusariviridae* family. The first BLASTX hit was the unpublished sequence of Heterobasidion parviporum fusarivirus 1 (HetpaFV1) from Finland (GenBank OR644501) with 92% identity. The HetFV1 genome sequence of nearly 7 kb contains one ORF comprising an RdRp conserved domain and a DEXH-box helicase domain (Fig. 1). The aa sequence motifs within the RdRp were highly conserved among selected fusariviruses; HetFV1 shared the most similarities with Phlebiopsis gigantea fusarivirus 1 (Fig. S4 available at https://doi.org/10.6084/m9.figshare.26503525). As *Heterobasidion* spp. and *P. gigantea* often share the same ecological niche, conifer stumps, this observation brings up the question of whether the horizontal transmission of the common ancestor of these fusariviruses could have occurred between the two fungal taxa. Within the helicase domain, the DEXH motif of HetFV1 was identical to that of certain selected members of *Fusariviridae* and *Flaviviridae* (Fig. S5 available at https://doi.org/10.6084/m9.figshare.26503525). Phylogenetically, HetFV1 clustered in the *Gammafusarivirus* genus with HetpaFV1 and fusariviruses of *Lentinula edodes* (70) and *P. gigantea* (71; Fig. 3).

## Tobamo-like virus

A contig showing moderate similarity to members of the *Virgaviridae* family, with Lentinula edodes tobamo-like virus 1 (70) as its first hit in BLASTX, was named Heterobasidion tobamo-like virus 1 (HetTlV1). It caused a low titer infection of a single host strain, the mean depth of coverage was merely 20 reads (Table 1). The first tobamo-like virus in *Heterobasidion* is monopartite, 12.6 kb long (Fig. 1). Its largest ORF was predicted to encode a replicase including methyltransferase and helicase domains. Varying degrees of conservation of aa sequence motifs were observed across *Virgaviridae*, but overall, HetTlV1 showed the highest similarity to Podosphaera prunicola tobamo-like virus within both domains (Fig. S6 and S7). The second ORF encodes the RdRp, which appeared slightly more conserved than the upstream regions. Taking into account all aa sequence motifs within the RdRp, HetTlV1 was most similar to Plasmopara viticola lesion-associated tobamo-like virus 1 (Fig. S8 available at https://doi.org/10.6084/m9.figshare.26503525). The third ORF resembles a MP with DEAD-like helicases superfamily and helicase conserved C-terminal domains (Fig. 1). The last two ORFs are of unknown function. The number of nt between the ORFs encoding the replicase and the RdRp suggests that −1 ribosomal frameshifting is utilized to express these proteins. Similarly, +1 ribosomal frameshift appears to take place between the last two ORFs encoding HPs. The mechanism behind these frameshifts is unclear as neither typical shifty heptamer motifs nor octanucleotides promoting programmed ribosomal frameshifting (73) were noticed at the ends of the upstream ORFs. However, the first 200 nt of the downstream ORFs contain the UUAUG sequence, which is deemed to be a key motif for translation (65). Programmed −1 ribosomal frameshift has so far been described in one tobamovirus species, *Hibiscus latent Singapore virus*, whose hepta-adenosine stretch

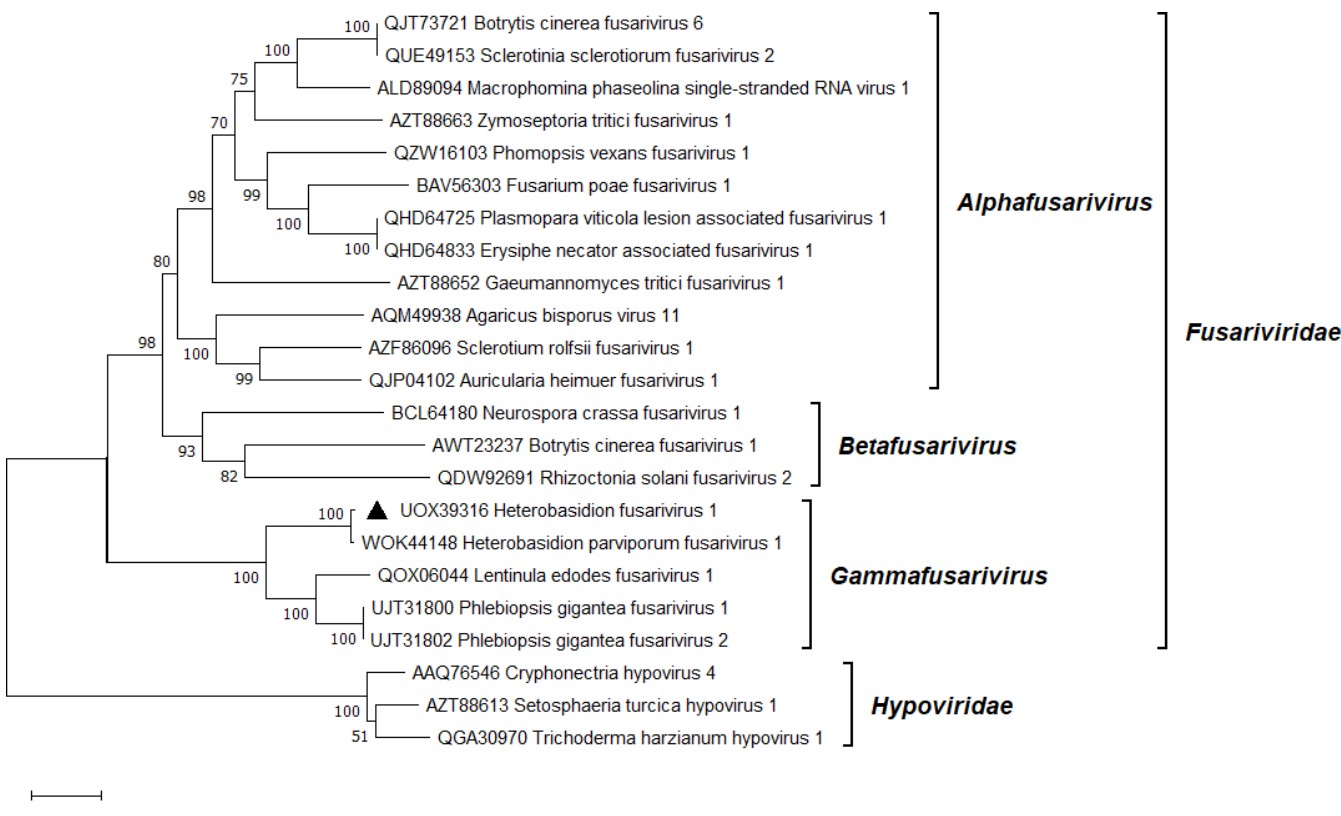

**FIG 3** Phylogenetic relationships of HetFV1 (denoted by a triangle) with selected members of the *Fusariviridae* family. The *Hypoviridae* family served as outgroup. The tree was built based on the alignment of RdRp aa sequences generated with MUSCLE in MEGA11 (54). The evolutionary history was inferred by using the maximum likelihood method and Le_Gascuel_2008 model (72). A discrete Gamma distribution was used to model evolutionary rate differences among sites (five categories + *G* + *I*). All positions with less than 95% site coverage were eliminated, that is, fewer than 5% alignment gaps, missing data, and ambiguous bases were allowed at any position. Evolutionary analyses were conducted in MEGA11 with 1,000 bootstrap repeats. The percentage of trees in which the associated taxa clustered together is shown next to the branches. Branch lengths are proportional to the number of substitutions per site. GenBank accessions are displayed before virus names.

located in its replicase gene triggers the frameshift (74). Phylogenetic analysis placed HetTlV1 as a separate individual branch at a large distance from classified and unclassified members of *Virgaviridae* (Fig. 4).

## Ambi-like viruses

Ambiviruses constitute a recently discovered but rapidly expanding group of mycoviruses classified as a new phylum by the International Committee on Taxonomy of Viruses (ICTV) (26, 76). Their distinctive features include the possession of two non-overlapping ORFs in ambisense orientation (77) and the activity of self-cleaving ribozymes enabling rolling circle replication (51). The remarkable abundance of ambi-like viruses has been reported via data mining in metatranscriptomes (51, 78) and fungal transcriptomes (79). The present study was no exception. One of the detected contigs shared 96% BLASTX identity with the virus strain HetAlV3-pa1 known from Finland (23; Table 1). The 3′ terminus of the HetAlV3 genome sequence was completed by RACE based on three cloned amplicons spanning 1,269 bp. Notably, the read coverage of HetAlV3 was markedly low both in the study by (23) and in the present work. Apart from HetAlV3, 10 new ambi-like viruses were retrieved from our pooled data sets, designated as HetAlV5–9, HetAlV24, and HetAlV26–29 (Table 1). The RNA-seq analysis of *H. annosum* strain 1987 revealed the existence of six further ambi-like viruses, named HetAlV19–23 and HetAlV25 (Table 1). These originally went undetected in the ANNO1 data set, but their genome

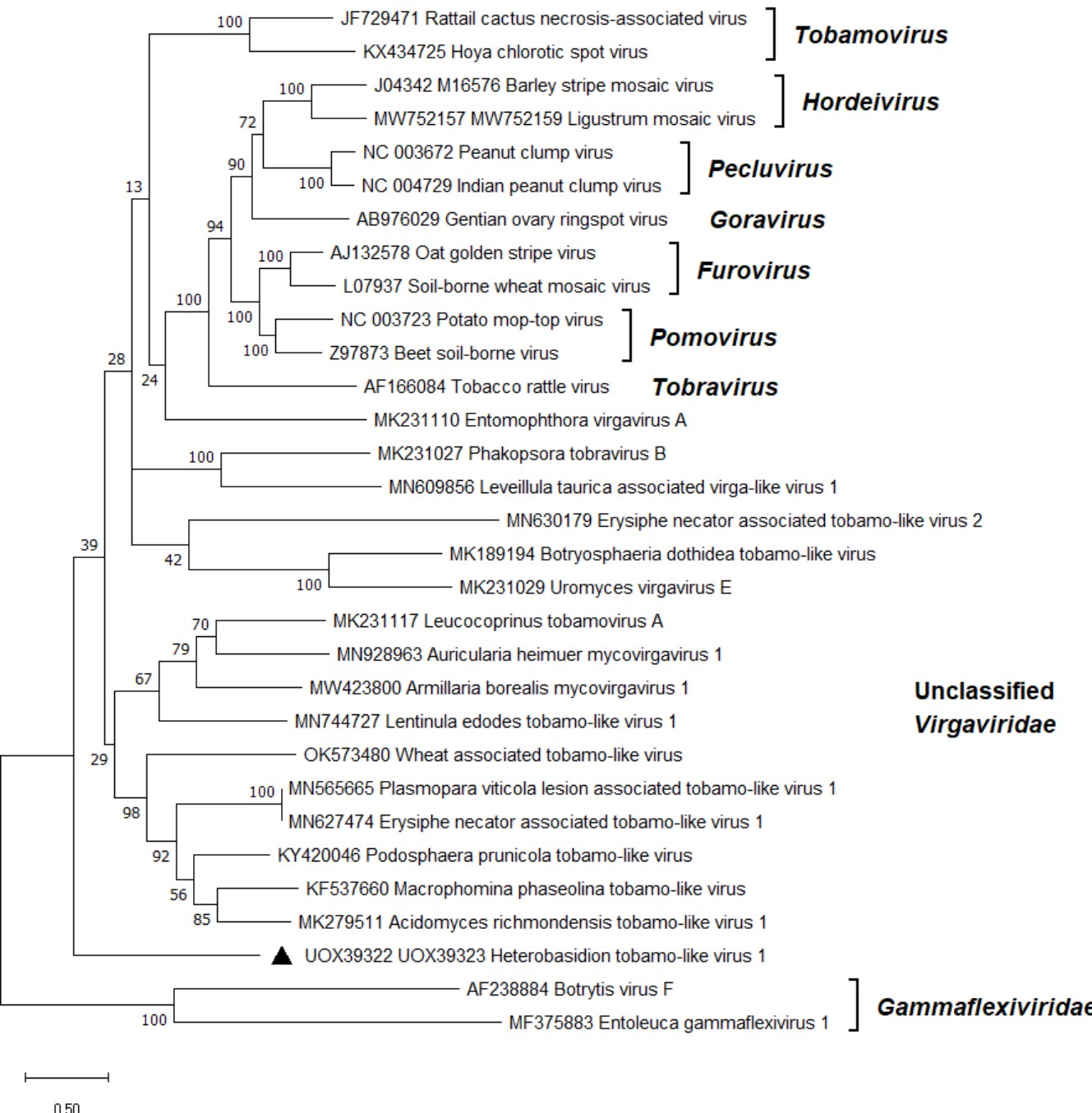

**FIG 4** Phylogenetic relationships of HetTlV1 (denoted by a triangle) with selected members of the *Virgaviridae* family. The *Gammaflexiviridae* family served as outgroup. The tree was built based on the codon-aligned nt sequences of the replication proteins and RdRps generated with MUSCLE in MEGA11 (54). The evolutionary history was inferred by using the maximum likelihood method and Kimura 2-parameter model (75). A discrete Gamma distribution was used to model evolutionary rate differences among sites (five categories + $G$ + $I$). All positions with less than 95% site coverage were eliminated, that is, fewer than 5% alignment gaps, missing data, and ambiguous bases were allowed at any position. Evolutionary analyses were conducted in MEGA11 with 500 bootstrap repeats. The percentage of trees in which the associated taxa clustered together is shown next to the branches. Branch lengths are proportional to the number of substitutions per site. GenBank accessions are displayed before virus names.

sequences were posteriorly shown to have varying levels of read support in both data sets (Table 1). A second RT-PCR was needed to obtain a well-visible amplicon within the RdRp encoding ORF of HetAlV23 and HetAlV29, both of which had low read coverage, as well as that of HetAlV6 from isolate 2038. Most likely, HetAlV6 caused a low titer infection

in this particular isolate as it showed a moderate read coverage while being present in five *H. annosum* strains altogether (Table 1). According to BLASTX search, the newly discovered ambi-like viruses were 33%–78% identical to known ambi-like viruses of *Heterobasidion* or *Armillaria* (Table 1). As usual for ambiviruses, each virus reported here had a genome of approximately 5 kb, comprising two bidirectional ORFs. The contigs resembling ambi-like viruses were almost always assembled with the putative RdRp-encoding ORF in the sense orientation. Contigs assembled in the reverse or both orientations were reverse complemented, and the final genomic sequences were deposited into GenBank with the putative RdRp in the coding orientation. In several cases, the ambiviral contigs were assembled as dimers or trimers of the same genomic sequence. This observation is consistent with previous studies (77, 80), indicating that ambiviruses have circular genomes (26). Each ambi-like virus contained at least one hairpin (HPRz) or hammerhead (HHRz) ribozyme, typically positioned after the C-termini of both ORFs (Table S3 available at https://doi.org/10.6084/m9.figshare.26503525). The genome organization of HetAlV6, representative of the novel ambi-like viruses, is depicted in Fig. 1. For HetAlV28 and HetAlV29, only a partial genome was determined, lacking either the RdRp (ORFA) or the HP (ORFB). No conserved domains were found, but the longest predicted protein of each virus with a complete genome included the GDD motif, which is considered the hallmark of RdRps. Virus candidates showing >90% nt pw identity over their putative RdRp encoding region were considered strains of the same virus species. This is an arbitrary threshold/species demarcation criterion suggested by the ICTV experts based on existing data (26). Pw identities of RT-PCR amplicons of ambi-like virus strains ranged 88%–97% at the nt level and 95%–100% at the aa level (Fig. S3 available at https://doi.org/10.6084/m9.figshare.26503525). The similarity of RdRps was generally slightly higher than that of HPs. The ambi-like viruses of *Heterobasidion* are presumably of polyphyletic origin as they were placed in three separate branches (Fig. 5). The majority belong to the *Trimbiviridae* family, a smaller group to *Dumbiviridae*, whereas HetAlV6 fell into *Quambiviridae*, closely associated with Armillaria novae-zelandiae ambi-like virus 1 (81). With two exceptions (isolates 1991 and 2040), the *Heterobasidion* ambi-like viruses were in coinfections in their host strains with CVMPlS, HetFV1, ambi-like, narna-like, or ourmia-like viruses (Table 1). HetAlV1–4 were always found in mixed infections in Finnish *H. parviporum* isolates (23). Ambi-like viruses of other basidiomycetes, such as *Tulasnella* spp. (77), *Armillaria* spp. (81), and *P. gigantea* (71) were shown to be responsible for both single and multiple virus infections. On the other hand, Cerato-basidium ambivirus 1 (77) and Cryphonectria parasitica ambivirus 1 (80) were detected in single infections.

## Coguvirus movement protein-like sequence

The cDNA libraries ANNO1 and 1987 yielded a contig with resemblance to the MP encoding segment of plant coguviruses. The contig was most similar (25% identity in BLASTX) to the putative MP of the unclassified Jiangsu sediment phenui-like virus (82). Putative segments encoding for the polymerase and the nucleocapsid protein were absent in the data, only a 2.5 kb long segment of the positive-sense antigenome was assembled. The genomic segment encompasses a polyT tail of 109 nt and encodes a MP (Fig. 1). DNA analysis indicated that the sequence was not genome-integrated. The fragment was designated as Coguvirus movement protein-like sequence (CVMPlS; Table 1) and represents the first putative negative-strand RNA virus in *Heterobasidion*. CVMPlS seems to be distantly related to classified members of the *Phenuiviridae* family (Fig. 6), although a tree built on the MP sequence may only give an approximate idea of its phylogenetic position. CVMPlS infected the multivirus isolate 1987 together with a fusarivirus, ambi-like virus, and narna-like virus (Table 1).

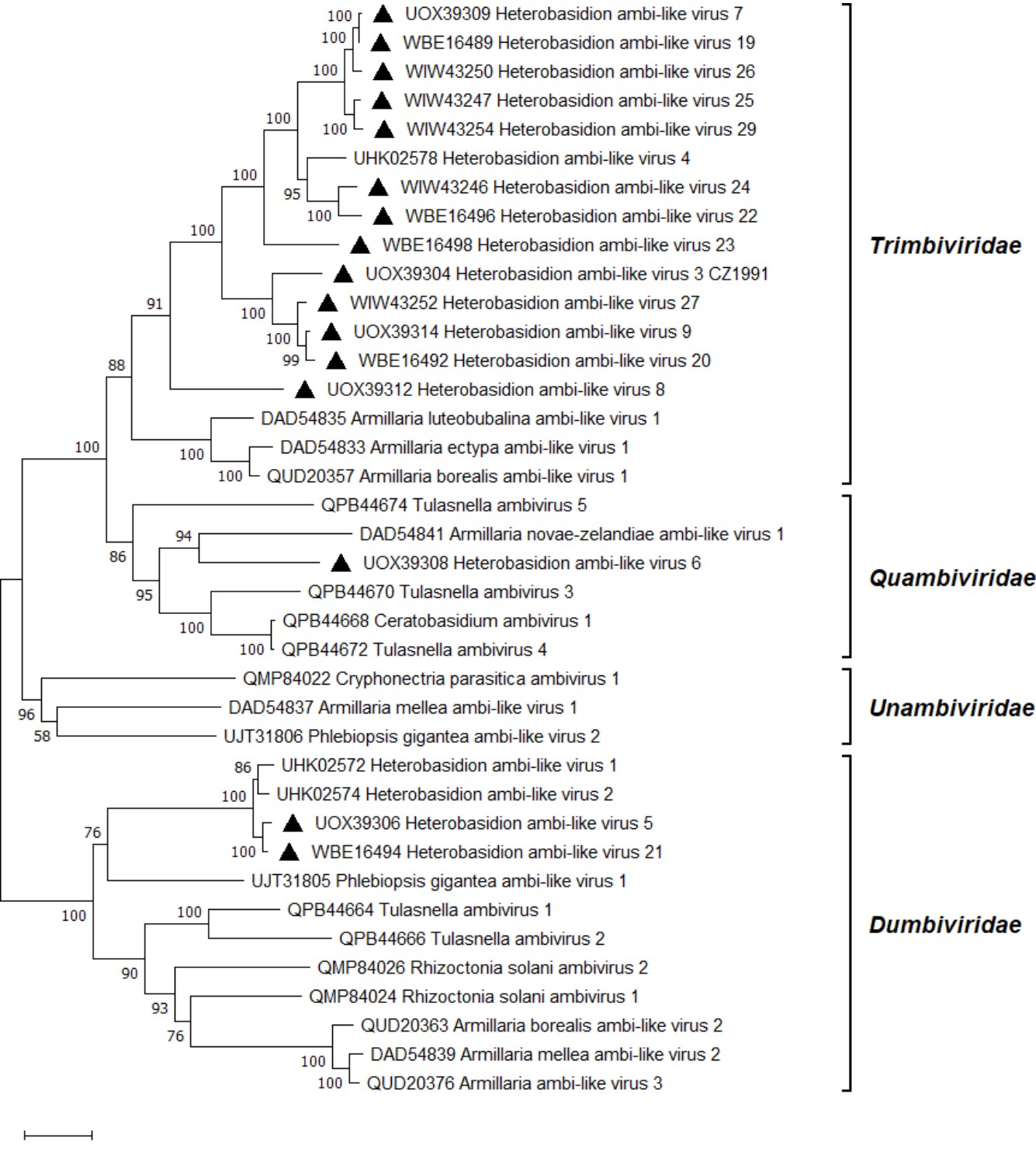

**FIG 5** Phylogenetic relationships of ambi-like viruses. The tree was built based on the alignment of RdRp aa sequences generated with MUSCLE in MEGA11 (54). The evolutionary history was inferred by using the maximum likelihood method and Le_Gascuel_2008 model (72). A discrete Gamma distribution was used to model evolutionary rate differences among sites (five categories + G + I). All positions with less than 95% site coverage were eliminated, that is, fewer than 5% alignment gaps, missing data, and ambiguous bases were allowed at any position. Evolutionary analyses were conducted in MEGA11 with 1000 bootstrap repeats. The percentage of trees in which the associated taxa clustered together is shown next to the branches. Branch lengths are proportional to the number of substitutions per site. GenBank accessions are displayed before virus names. Viruses described in this study are denoted by a triangle.

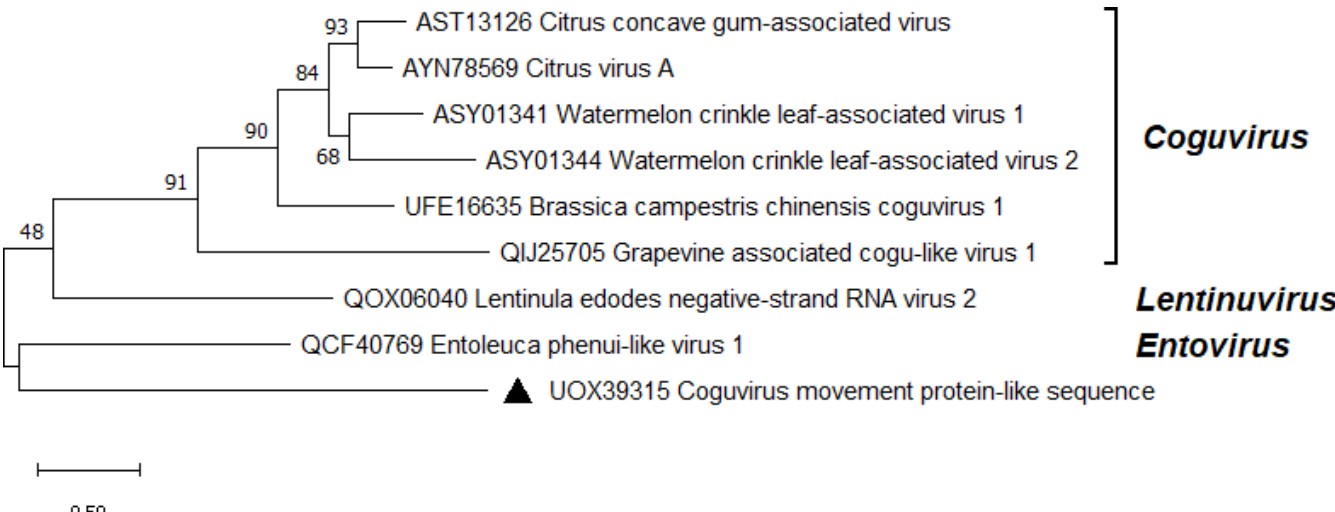

**FIG 6** Phylogenetic relationships of CVMPlS (denoted by a triangle) with selected members of the *Phenuiviridae* family. The tree was built based on the alignment of MP aa sequences generated with MUSCLE in MEGA11 (54). The evolutionary history was inferred by using the maximum likelihood method and Le_Gascuel_2008 model (72). A discrete Gamma distribution was used to model evolutionary rate differences among sites (five categories + *G*). All positions with less than 95% site coverage were eliminated, that is, fewer than 5% alignment gaps, missing data, and ambiguous bases were allowed at any position. Evolutionary analyses were conducted in MEGA11 with 1,000 bootstrap repeats. The percentage of trees in which the associated taxa clustered together is shown next to the branches. Branch lengths are proportional to the number of substitutions per site. GenBank accessions are displayed before virus names.

## Efficiency of virus elimination and horizontal transmission

By the time we commenced the virus removal (curing) experiments, HetAlV23 had already been lost from isolate 1987 during its storage at 4°C for 1 year. This could be explained by its extremely low accumulation level, a mean depth of 12 reads (Table 1). Viruses were retained in the monohyphal cultures in widely varying frequencies (0%–100%). HetTlV1 was absent from the hyphal tip isolates of strain 1989. Contrastingly, HetOlV2 was transmitted to 10 of 10 monohyphal cultures of strain 2072. In strain 1987, which was infected by multiple viruses, the success rates of virus elimination were more nuanced. HetAlV20–22 were retained in all cases, HetNlV4 in 90%, HetFV1 in 70%, and HetNlV2 in 20% of the monohyphal cultures, whereas HetAlV19, HetAlV25, and CVMPlS were absent. Generally, viruses perceived to cause infections of low titer based on their read coverage were less stable. Still, characteristics of the virus and/or the fungal genotype also seemed to influence the likelihood of transmission. These findings are consistent with the results of (29), who found that the higher concentration of dsRNA in an *H. parviporum* isolates increased the dsRNA transmission rate into single hyphal tip isolates.

The original isolates 1987, 1989, and 2072, as well as the monohyphal cultures 1987–A2, –A4, –A5, and –C1, containing different combinations of viruses (Fig. 7A), were fully cured of their viral infections after both 1 and 2 weeks of thermal treatment. Temperature treatment lasting 2 or 4 weeks at 32–33°C was previously successfully applied to cure *Heterobasidion* strains of partitiviruses (14, 28), but even lower temperatures have been shown efficient: a 2-week incubation at 28°C resulted in loss of the dsRNA genomes of partitiviruses and curvulaviruses in *H. parviporum* (83).

After the maintenance of dual cultures for 1 month, horizontal transmission of virus(es) occurred in just one of nine pairings. Isolate 2052 received HetAlV19 and HetNlV4 from the donor strain 1987 when sampled from spot 1 (Fig. S2A and 7A). Interestingly, two additional viruses, HetAlV22 and HetNlV2, were transmitted to 2052 when sampled from spot 2, which was located farther from the interaction zone (Fig. S2A and 7A). Three months post-inoculation, 2052 received the same four viruses from 1987 as 1 month post-inoculation when sampled from spots 2 and 3 (Fig. S2B). However, the sample from spot 1 was infected by a different combination of viruses, that is, HetAlV21

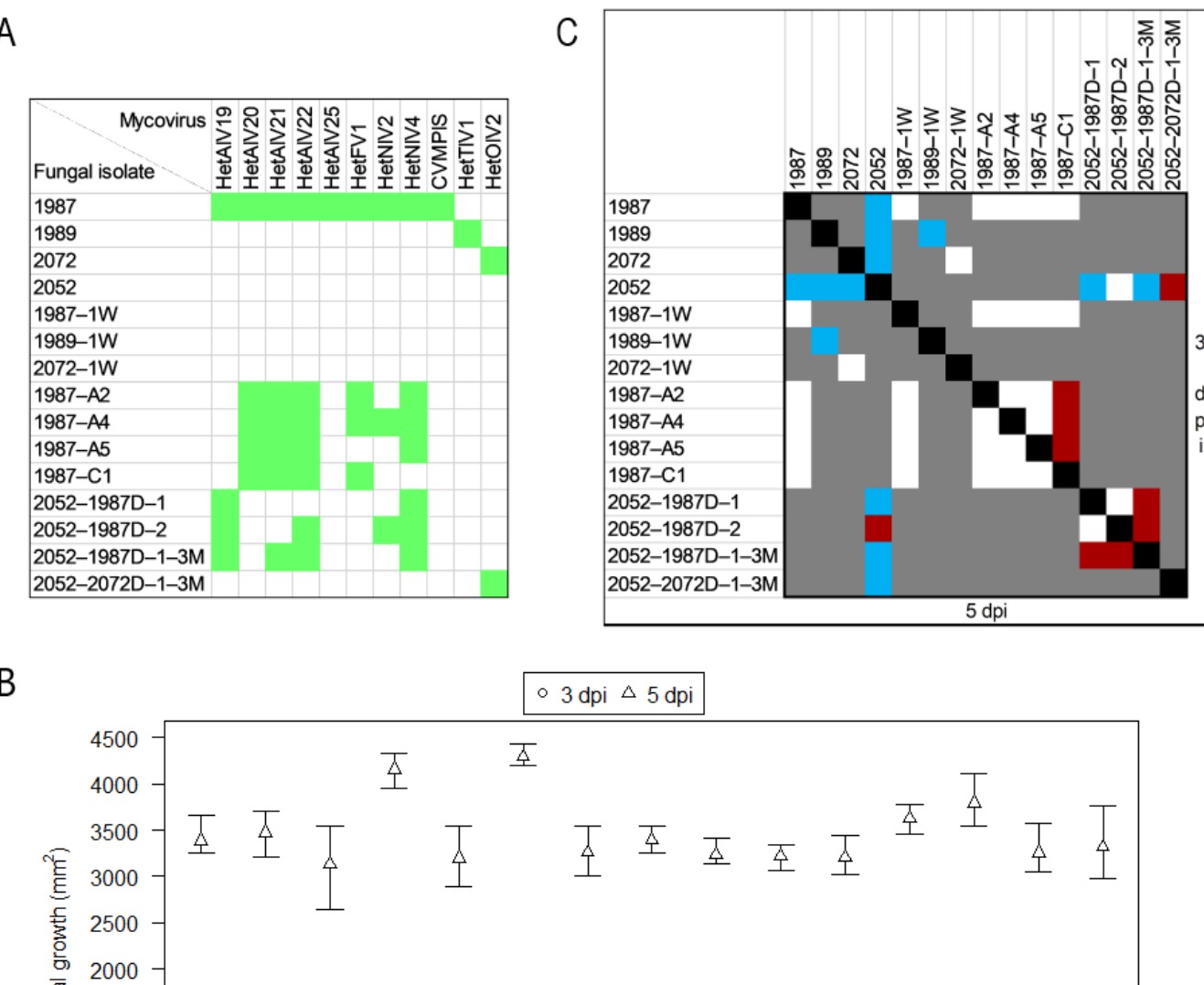

**FIG 7** Virus content influences the *in vitro* hyphal growth rate of *Heterobasidion*. (A) Presence of viruses in the tested *H. annosum* isolates. 1987, 1989, 2072, and 2052 denote original isolates with unaltered virus content. Note that HetAlV23 had been lost from 1987 during its storage at 4°C for 1 year. –1W, isogenic isolates fully cured of viral infections by thermal treatment lasting 1 week; –A2, –A4, –A5, –C1, monohyphal cultures created by hyphal tip isolation; 2052–, isogenic isolates, which have received different combinations of viruses by horizontal transmission from a donor strain denoted by D; –3M, dual cultures were maintained for 3 months (where absent, the isolation was performed after 1 month). (B) Mean area [mm²] of 2% malt extract agar plates covered by mycelium at 3 and 5 days post-inoculation (dpi), used as the metric to compare the fitness of *a priori* determined pairs of isolates. The bars represent the range of values of five technical replicates. (C) Statistical significance of differences in the growth rate between pairs of isolates, calculated with *t*-test. The upper right triangle includes comparisons at 3 dpi, and the bottom left triangle pertains to 5 dpi. Blue, $P < 0.001$; red, $P < 0.05$; white, $P > 0.05$; gray, not tested.

replaced HetNlV2 (Fig. S2B and 7A). Additionally, HetOlV2 was transmitted to 2052 from 2072 at sampling spots 1 (Fig. 7A) and 2. Based on RAMS fingerprinting (Fig. S9), the genotype of 2052 remained unaltered in all the above cases, which excluded the possibility of the viruses' presence due to intermixture of recipient and donor hyphae. The transmission of virus(es) was detected in nine other samples, but RAMS analysis indicated that these were contaminated with hyphae of the donor strain, which could be explained by the higher chance of intermingling and overgrowth of hyphae on the 3-month time horizon. Overall, the horizontal transmission rates of the investigated ssRNA viruses between conspecific *Heterobasidion* isolates were 0%–33%, which is rather low compared with previously observed rates for dsRNA viruses. Partitiviruses were success-fully transmitted in 74% of all tested donor-recipient pairings of *H. parviporum* (29, 31, 32); the transmission frequency of individual viruses ranged 0%–100%. The transmission rate of the curvulavirus HetRV6 between *H. parviporum* isolates was 43% (31). The alphapartitivirus HetPV13-an1 was transmitted from its natural host *H. annosum* to 69% of recipient *H. annosum* strains, but the two mitoviruses present in the donor were not transmitted at all (14). The transmission of viruses related to those hosted by our donor isolates has not been studied in *Heterobasidion in vitro*, although there is indirect evidence that ambi-like viruses are laterally transmitted between *H. parviporum* strains under natural conditions (23).

## Effects of virus infections on host growth

Results of the growth rate comparisons are presented in Fig. 7B and C. The naturally virus-free isolate 2,052 grew significantly faster than the original virus-hosting isolates 1987, 1989, and 2072 at both 3 and 5 dpi (all $P < 0.001$). These differences are notewor-thy but not necessarily indicative of any mycoviral effects. In fact, 2052 maintained its entire advantage when compared with the cured strains 1987–1W and 2072–1W but performed equally well as 1989–1W (Fig. 7B), which points to the role of inherent phenotypic differences between the tested isolates. In line with the above, no significant differences were found in the growth rates of 1987 or 2072 and their isogenic virus-free counterparts, but 1989–1W advanced significantly faster than 1989 at both points in time ($P < 0.001$). The presence of HetTlV1 caused a 19% growth reduction in its host at 5 dpi. This reduction indicates that this putative tobamovirus may be affecting essential processes within the host.

No significant differences were observed between 1987 or 1987–1W and the monohyphal cultures. Among the single hyphal cultures, 1987–C1 grew significantly slower than 1987–A2, –A4, and –A5 at 3 dpi ($P < 0.05$); however, differences were non-significant at 5 dpi. In three comparisons, it was possible to consider the effect of a single virus. Comparing 1987–C1 with –A2, we conclude that HetNlV4 may have a minor positive effect (12%) on host growth, which nonetheless diminished over time (Fig. 7). Confronting 1987–A2 with –A4 and –A5, we suppose that HetNlV2 and HetFV1 are cryptic. Latency is more common among narnaviruses than fusariviruses. Some members of *Fusariviridae* have been associated with hypovirulence in plant pathogenic fungi (84, 85).

The newly infected strain 2052–2072D–1–3M performed significantly worse than its isogenic counterpart at 3 dpi ($P < 0.05$) and 5 dpi ($P < 0.001$). Thus, the introduced HetOlV2 reduced the growth of its new host strain by 20%, despite not affecting its original host strain. The differential response of the two host strains to HetOlV2 indicates that the effects of the virus may be influenced by host-specific factors, such as genetic background, physiological differences, or differences in RNAi activity. In such cases, the virus could replicate more efficiently or evade host defenses in the new host strain, leading to reduced growth or other adverse effects (24, 86–89). As stated by (36), summarizing data on dsRNA viruses of *Heterobasidion*, "a single mycovirus strain may confer different effects on different host strains." Members of *Magoulivirus* are considered to establish latent infections with the exception of Fusarium oxysporum ourmia-like virus 1, which exhibited hypovirulence in inoculation experiments on bitter gourd plants (90).

The other isogenic isolates of 2052, infected by various combinations of ambi- and narna-like viruses, also showed some degree of underperformance. 2052–1987D–1 and 2052–1987D–1–3M were negatively affected at both 3 and 5 dpi (all $P < 0.001$), with a 5-dpi growth decrease of 13% and 22%, respectively. 2052–1987D–2 suffered a 9% growth reduction at 5 dpi ($P < 0.05$). Among these three derivatives of 2052,–1987D–1–3M grew significantly slower compared with the other two both at 3 and 5 dpi (all $P < 0.05$). These data imply that HetAlV19 and HetNlV4 may be somewhat detrimental in coinfection, and their effect is modified by further viruses present in the mycelium. The phenotypic effects of ambiviruses have been investigated in one study (81), where the ambi- and ourmia-like viruses of *Armillaria* spp. did not have a major effect on the laboratory growth rate of their hosts either as single or coinfections. Sclerotinia sclerotiorum narnavirus 5 was shown to induce hypovirulence in its host strain harboring five other (+)ssRNA viruses (91), which accords with the present findings. As a final point, it should be emphasized that the taxonomic position of a mycovirus is not indicative of its effect on the host (19).

Taken together, although some differences were statistically highly significant, the extent of fungal growth reduction was modest across all the comparisons. For reference, HetPV13-an1 reduced the growth of different *H. annosum* and *H. parviporum* isolates by 82%–96% under similar experimental conditions (14, 30, 31). HetPV15-pa1 caused 80%–88% growth decrease in *H. annosum* (30). Other partitiviruses and HetRV6 were cryptic or had modest or non-significant effects, either positive or negative to *Heterobasidion* (22, 28, 30, 31, 36). (30) and (31) have shown that coinfecting dsRNA viruses of *Heterobasidion* affect each other's phenotypic effects on their hosts in an unpredictable manner, that is, the viral effects are not additive. The lack of any consistent trend for the isolates with multiple infections allows us to speculate that the same holds true for other taxonomic groups of *Heterobasidion* viruses. Detailed studies investigating the effects of each discovered virus in single and coinfections on multiple hosts would be the essential next steps for evaluating their suitability for potential virocontrol applications.

## Effects of virus infections on the host proteome profile

The proteomic analyses, utilizing *Heterobasidion annosum* v2.0 (59) proteome annotations, resulted in the identification and quantification of 2357 and 1660 protein families, respectively (Table S4 available at https://doi.org/10.6084/m9.figshare.26503525). The experiment failed to identify any viral proteins. A low-confidence search focusing only on expected viral proteins yielded few potential matches, but these were also found in samples without viruses and were likely false positives. That suggests that either the virus produces very few proteins or the fungus actively suppresses viral transcription/translation.

Although no viral proteins were found in the proteome analysis, a comparison of isolates containing viruses (1987, 1989, and 2072) with a virus-free isolate (2052) revealed 510 proteins with statistically significant differences in abundance (differentially abundant proteins, DAPs; $P < 0.05$). These DAPs constituted over 40% of the estimated protein levels in isolate 2052. Additionally, the corresponding principal component analysis (PCA) showed a clear distinction between the isolates, with the first principal component (representing 43.4% of the variation) appearing to separate samples based on their viral load (Fig. 8A). The pw comparisons showed relatively low overlap in identified DAPs ($t$-test, $P < 0.05$; relative fold change >1.4), and only 78 proteins showed a similar response to virus presence (Fig. 8B). The subset of proteins that demonstrated an increase in abundance included three enzymes involved in detoxification and stress response (heme peroxidase, DyP-type peroxidase and 2-oxoglutarate/Fe(II)-dependent dioxygenase), proteins participating in targeted protein degradation (DUF431-domain-containing protein and two ubiquitin-conjugating enzymes), various carbohydrate-active enzymes (CAZymes), and an RNA-binding protein (RRM domain-containing protein). In all three virus-infected isolates, an enzyme laccase showed a significant increase in abundance. This finding aligns with prior observations

of viral influence on fungal laccase production (92). Proteins that possibly participated in the fungal response to viruses and showed a significant decrease in abundance included (among others) nine isoforms of glycoside hydrolases, which are believed to play a role in biotic interactions (93), two protein kinases, PKS_ER domain-containing proteins that are involved in secondary metabolism, cytochrome P450 monooxygenases, an enzyme in chitin biosynthesis glutamine-fructose-6-phosphate aminotransferase, proteases, and several proteins with an unknown function.

In the second set of comparisons, the impact of virus introduction was observed in isolate 2052. Newly infected isolates displayed only 139 DAPs based on ANOVA and Tukey's HSD test. However, the corresponding PCA successfully grouped the infected isolates as expected (Fig. 8C). Further analysis using consecutive pw comparisons revealed 97 DAPs ($t$-test, $P < 0.05$; relative fold change >1.4; Fig. 8D) with a similar profile in at least three different isolates. Among the DAPs consistently accumulated in all infected isolates were candidates of interest, including a putative ortholog of Prd-6, an ATP-dependent RNA helicase involved in nonsense-mediated mRNA decay, and two proteins involved in ribosome biogenesis: an ATP-dependent RNA helicase involved in the biogenesis of 40S and 60S ribosomal subunits and a ribosomal 60S subunit protein L30. Ribosomal proteins are now a well-established part of fungi–virus interaction (95), and nonsense-mediated mRNA decay has been found to restrict virus replication in both plants and animals (96). DAPs with higher abundance in at least three infected isolates included two cytochrome P450 monooxygenases, an F-box domain-containing protein (involved in targeted protein degradation), and an RRM domain-containing protein (RNA binding). It should be noted that among the 97 filtered DAPs, 72 were also present in the first data set. Notably, the response to viral presence was similar for 60 of these shared DAPs.

Finally, the proteomes of cured isolates and their corresponding infected genotypes were compared. Despite the relatively large sets of DAPs, only 12 DAPs were found in more than one comparison (Table S4 available at https://doi.org/10.6084/m9.figshare.26503525), and only seven of these showed a similar trend. Two dioxygenases (Isopenicillin N synthase-like and Fe2OG dioxygenase domain-containing protein) showed significantly higher abundance in cured isolates. The same was found for a nuclear cap-binding protein subunit and a putative autophagy-associated protein (ATG16). Conversely, a lipid metabolism enzyme (3-oxoacyl reductase), an adhesin domain-containing protein, and pyruvate decarboxylase exhibited increased abundance in virus-containing isolates (Table S4 available at https://doi.org/10.6084/m9.figshare.26503525). The comparison of partially cured isolates 1987 also showed very high diversity with only 14 DAPs showing a similar trend in more than two partially cured isolates (Table S4 available at https://doi.org/10.6084/m9.figshare.26503525). Besides already listed proteins, this data set identified the putative role of two additional proteins: a Hydantoinase_B domain-containing protein (showing a significant increase in abundance in cured isolates) and a cytosolic Ca2+-dependent cysteine protease (more abundant in infected isolates) that could participate in viral protein degradation.

Taken together, this analysis identified over 500 protein candidates encompassing diverse metabolic and signaling pathways as potential viral response proteins. Approximately 150 of these candidates exhibited the expected trends in abundance changes across at least two of the analyses (Table S4 available at https://doi.org/10.6084/m9.figshare.26503525).

## Conclusions

Our findings indicate that the virome of *Heterobasidion* populations in Czechia is highly diverse and seems to differ from that in the boreal region. Notably, the most frequently reported virus of the pathogen, HetRV6 was not found among 85 *Heterobasidion* strains from Czechia or Slovakia. Earlier, HetRV6 has been detected in the neighboring countries Austria and Poland (22). Likewise, the *Partitiviridae* family, with 21 members described to date from *Heterobasidion*, was not represented in our data sets, although present in the

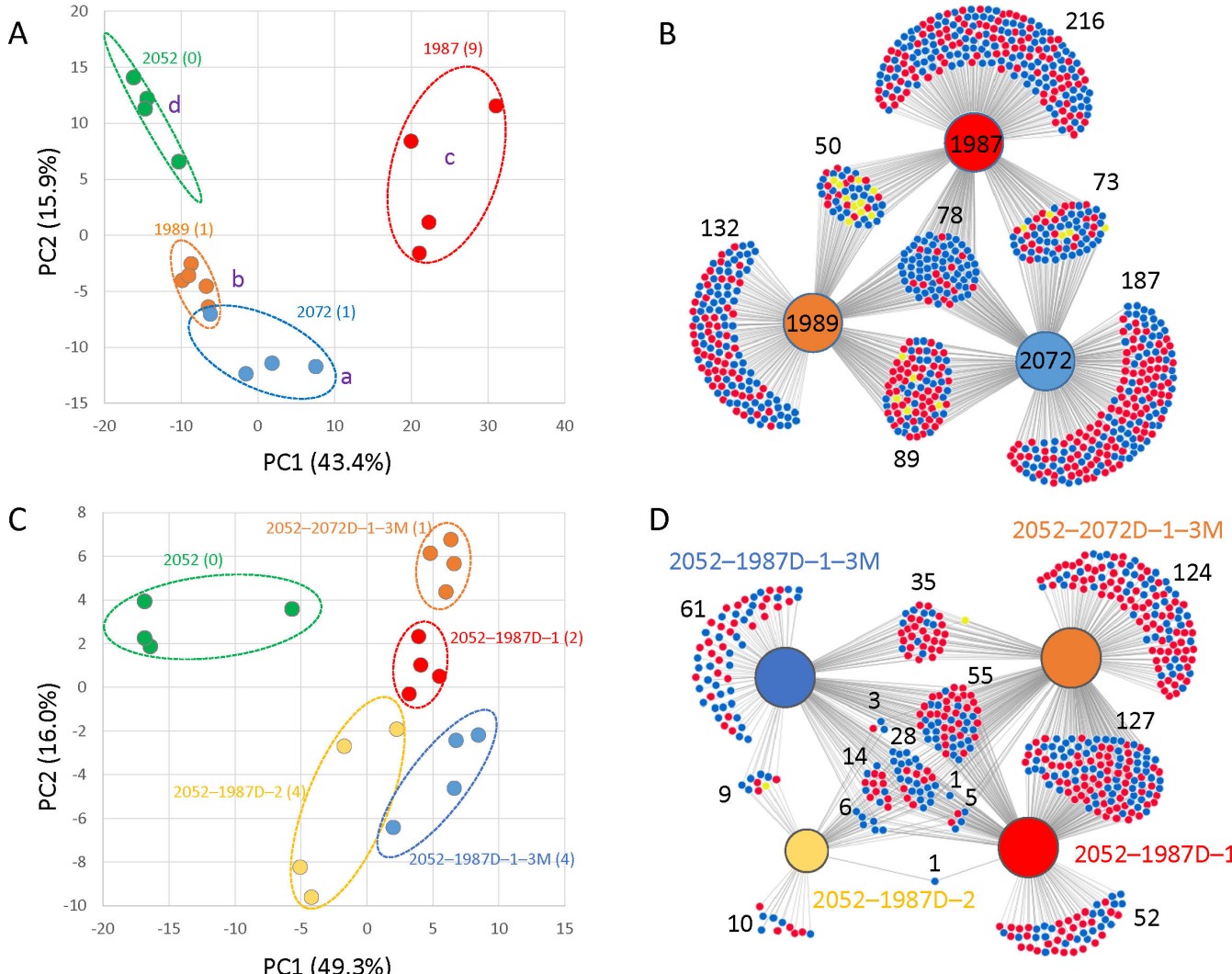

**FIG 8** *Heterobasidion* proteome profile reveals distinct signatures in virus-free and infected mycelia. (A) Principal component analysis (PCA) based on 510 differentially abundant proteins (DAPs) identified in four isolates with varying viral content (ANOVA, $P < 0.05$, Tukey's HSD). Different letters indicate statistically significant differences between groups (Conover's test, $P < 0.05$). Numbers in brackets represent the number of different viruses detected in each isolate. (B) Pairwise comparisons of individual isolates to the virus-free isolate 2052, visualized using a DiVenn diagram (94). Proteins with a statistically significant change in abundance compared to isolate 2052 (Student's *t*-test, $P < 0.05$; fold change >1.4) are represented by dots. Red dots indicate proteins with increased abundance, and blue dots indicate proteins with decreased abundance in the compared isolate relative to 2052. Yellow dots represent proteins that showed contrasting responses (increased in one comparison, decreased in another) between isolate pairs. (C-D) Impact of viral infection on the proteome profile: PCA based on all 139 identified DAPs (ANOVA, $P < 0.05$, Tukey's HSD), and the results of pairwise comparisons visualized using a DiVenn diagram ($P < 0.05$; fold change >1.4). All results are based on four biological replicates. The virus content of each isolate is shown in Fig. 7A. For detailed information on protein identifications and statistical analyses, see Table S4 available at https://doi.org/10.6084/m9.figshare.26503525.

neighboring Poland. However, two previously known and 23 novel ssRNA mycoviruses were discovered. We report the first putative negative-strand RNA virus in *Heterobasidion*. RNA-seq of a single isolate had clear benefits over RNA-seq of pooled libraries, such as the discovery of new ambi-like viruses and a narnaviral segment, the completed genome of a virus causing low titer infection, and resolved ambiguous bases in certain viral contigs. Some of the tested viruses exhibited a negative impact on the mycelial growth rate and induced considerable alterations in the host proteome profile. Considering both the growth rate test and proteomics, the candidates deemed most suitable for further studies are HetTlV1 and HetOlV2. More extensive studies are needed to evaluate their biocontrol potential.

## ACKNOWLEDGMENTS

Computational resources were supplied by the project "e-Infrastruktura CZ" (e-INFRA CZ LM2018140) supported by the Ministry of Education, Youth and Sports of the Czech Republic. We would like to thank Milica Raco (Oregon State University, Corvallis, United States) for guidance in RNA extractions and bioinformatics. We are thankful to Ondřej Hejna (Department of Genetics and Agrobiotechnology, University of South Bohemia in České Budějovice) for support in bioinformatic analysis. Petr Sedlák, Michal Tomšovský, Libor Jankovský and Miloň Dvořák (all Department of Forest Protection and Wildlife Management, Mendel University in Brno) are gratefully acknowledged for invaluable assistance during the collection of Heterobasidion isolates.

This research was funded by the Specific University Research Fund of the Faculty of Forestry and Wood Technology, Mendel University in Brno, grant number LDF_VP_2019034. Additionally, it was supported by the European Regional Development Fund, Project "Phytophthora Research Centre", Reg. No. CZ.02.1.01/0.0/0.0/15_003/0000453, EC/HE/101087262/ERA-Chair:Striving for Excellence in the Forest Ecosystem Research/EXCELLENTIA and by Generalitat Valenciana through PROMETEO program, Project CIPROM/2022/21.

All authors contributed to the study conception and design. L.B.D., M.Č., and L.B. conducted experiments. L.B.D., M.Č., M.d.l.P., and L.B. analyzed data. L.B.D., M.Č., and L.B. wrote the manuscript and all authors commented on previous versions of the manuscript. All authors read and approved the manuscript.

## AUTHOR AFFILIATIONS

[1]Department of Forest Protection and Wildlife Management, Faculty of Forestry and Wood Technology, Mendel University in Brno, Brno, Czechia
[2]Department of Molecular Biology and Radiobiology, Faculty of AgriSciences, Mendel University in Brno, Brno, Czechia
[3]Instituto de Biología Molecular y Celular de Plantas, Universidad Politécnica de Valencia-CSIC, Valencia, Spain
[4]Natural Resources Institute Finland (Luke), Helsinki, Finland

## AUTHOR ORCIDs

László Benedek Dálya ⓘ http://orcid.org/0000-0002-9443-4832
Martin Černý ⓘ http://orcid.org/0000-0002-0651-4219
Marcos de la Peña ⓘ http://orcid.org/0000-0002-7949-8459
Anna Poimala ⓘ http://orcid.org/0000-0001-9374-3783
Eeva J. Vainio ⓘ http://orcid.org/0000-0002-6739-7968
Jarkko Hantula ⓘ http://orcid.org/0000-0002-1016-0636
Leticia Botella ⓘ http://orcid.org/0000-0002-6613-5405

## FUNDING

| Funder | Grant(s) | Author(s) |
|---|---|---|
| Mendelova Univerzita v Brně (MENDELU) | LDF_VP_2019034 | László Benedek Dálya |
| | | Leticia Botella |
| EC \| European Regional Development Fund (ERDF) | CZ.02.1.01/0.0/0.0/15_003/0000453 | László Benedek Dálya |
| | | Martin Černý |
| | | Leticia Botella |
| Generalitat Valenciana (GVA) | Project CIPROM/2022/21 | Marcos de la Peña |

| Funder | Grant(s) | Author(s) |
|---|---|---|
| | EC/HE/101087262/ERA-Chair:Striving for Excellence in the Forest Ecosystem Research/EXCELLENTIA also supported the publication of this article. | |

## DATA AVAILABILITY

The RNA-Seq data have been deposited into the NCBI Sequence Read Archive database with the accession numbers SRR18240564, SRR18240565, SRR18240566 and SRR18961509 under the BioProject PRJNA809936. The mycoviral genomic sequences are available from the NCBI GenBank. The mass spectrometry proteomics data have been deposited to the ProteomeXchange Consortium via the PRIDE (97) partner repository with the data set identifier PXD045991.

## ADDITIONAL FILES

The following material is available online.

### Open Peer Review

**PEER REVIEW HISTORY (review-history.pdf).** An accounting of the reviewer comments and feedback.

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
