## [Reviewer comments · mSystems]

Diversity and impact of single-stranded RNA viruses in Czech Heterobasidion populations

László Dályaa, Martin Cerny, Marcos de la Peña, Anna Poimala, Eeva Vainio, Jarkko Hantula, and Leticia BOTELLA

Corresponding Author(s): Leticia BOTELLA, Mendelova univerzita v Brne Lesnicka a drevarska fakulta

Review Timeline:

Submission Date:	April 6, 2024
Editorial Decision:	June 19, 2024
Revision Received:	July 27, 2024
Accepted:	August 4, 2024

Editor: Liyuan Ma

Reviewer(s): Disclosure of reviewer identity is with reference to reviewer comments included in decision letter(s). The following individuals involved in review of your submission have agreed to reveal their identity: İkbal Agah İnce (Reviewer #1)

Transaction Report:

DOI: <https://doi.org/10.1128/msystems.00506-24>

Re: mSystems00506-24 (Diversity and impact of single-stranded RNA viruses in Czech Heterobasidion populations)

Dear Dr. Leticia BOTELLA:

The figures are not ready for publication, please revise them accordingly.

Revision Guidelines

Sincerely,
Liyuan Ma
Editor
mSystems

Reviewer #1 (Comments for the Author):

General Comments:

The manuscript titled "Diversity and impact of single-stranded RNA viruses in Czech Heterobasidion populations" presents an interesting study on the virome of Heterobasidion populations. While the study offers valuable insights, certain aspects require clarification to enhance the overall readability and reproducibility of the findings.

Major Concerns:

The RNA-seq analysis methodology described in the manuscript is somewhat unclear. It is essential to provide a detailed step-by-step description of the RNA-seq workflow, including the following:

- The specific criteria used for quality control and filtering of raw reads.
- Detailed information about the bioinformatic pipelines used for virus discovery, including versions of software and specific parameters/settings.
- The criteria for selecting viral contigs and the subsequent steps for validation.
- An explicit explanation of how the pooling of RNA samples was handled and how it may affect the results.

The methodology section needs revision to include these details to ensure other researchers can replicate the study.

-The manuscript lacks a detailed description of the models used in the phylogenetic analysis. The selection of appropriate models is crucial for the reliability of phylogenetic inferences.

I recommend identifying the models used for phylogenetic analysis and providing a rationale for their selection. Additionally, it includes the software and versions used for constructing phylogenetic trees.

-The figure legends provided are insufficient for readers to understand the presented data fully. Each figure should be self-explanatory, providing enough context and detail about what is being shown.

The figure legends need to be elaborated to include explanations of symbols, scales, and annotations. Ensure that each figure can stand alone without requiring the reader to refer to the main text.

Specific Comments:

In Section 2.4 - Bioinformatics:

-It would be helpful to clarify the two different approaches combined for in silico data mining. Specify if these approaches were applied to all datasets or if there were variations.

In Table 1 - Virus Detection:

-Include additional columns in the table to specify the coverage and depth of each detected virus. This will help readers assess the reliability of the detection.

In Supplemental Figures:

-Ensure that all supplemental figures are referenced appropriately in the main text. It is currently unclear how some supplemental data supports the presented findings.

Reviewer #2 (Comments for the Author):

This study documents the mycoviruses inhabiting an important and impactful fungal pathogen of conifers in Czechia. The authors contextualize their mycoviral findings with expectations given the Heterobasidion viruses in other parts of Europe, and interestingly found very little similarity in viromes. The most unique, and perhaps novel, aspect of the study was the proteomics work, which is rarely done in mycovirus research but is an important future direction of research. While the results of the proteomic profiling may not have been what was expected since no viral proteins were detected, it is an important step towards understanding enigmatic fungal-viral interactions and the different abundances of metabolic proteins are interesting. Overall the authors make sound interpretations of the results across various experiments, and it is a worthy study. My primary suggestion is that the authors revisit the figures in order to make them publication-ready and restructure the figures to ensure the most important ones are primary and additional are supplemental. An additional map figure would add interesting context. Specific comments to follow:

Major:

Figure 1 should be supplemental.

Figure 2 It may be more useful to the reader to define the pfams. For instance, you could add a legend to define the pfams (05919 = mitovir pol, etc.) I'm sure there are additional ways to make this information more reader-available, too.

Figure 8 A- valuable information. B-What is this showing? Why is this important/what value do you see this adding to the paper?

C- I don't understand the comparison..... it looks like it is comparing the area of isolates grown at 5dpi to isolates grown at 3dpi... which doesn't make any sense.

Overall (fig 8):Can you find an alternative way to present these data? Maybe delta, or growth rate? The statistics reported are not able to be visualized by these figures (lines 589, 592-594, etc). Be sure to label axes in the new figure.

Figure 9 A & C- Would be more clear to use the traditional circling approach to demonstrate significant differences between groups. Would be more visually clear and appealing to have a legend with the isolates and their corresponding colors than have the text just near the data points as is.

Minor:

Line 21 strike extra use of word "conifer"

Lines 56-60 This left me feeling some conflict. Heterobasidion is a problem... but not a very big problem relevant to bark beetle?

Would it be accurate to articulate that the challenge lies in how bark beetle overshadows heterobasidion in terms of damage?

Or, is bark beetle SPREADING heterobasidion? Please clarify.

Line 504-505 Unclear what the "two bigger clusters" that are "grouped together" is referring to.

Line 507 I think this is just meaning because of their existence in hosts with additional infections there is now co-infection, but it reads as somehow the ambi viruses facilitate co-infection with multiple viruses. Please adjust for clarity.

General Have you consider any sort of spatial analyses? Even a map of sampling sites and viruses at each would be interesting; this map could also include results from other studies. Not required, but would add valuable context.

This study documents the mycoviruses inhabiting an important and impactful fungal pathogen of conifers in Czechia. The authors contextualize their mycoviral findings with expectations given the *Heterobasidion* viruses in other parts of Europe, and interestingly found very little similarity in viromes. The most unique, and perhaps novel, aspect of the study was the proteomics work, which is rarely done in mycovirus research but is an important future direction of research. While the results of the proteomic profiling may not have been what was expected since no viral proteins were detected, it is an important step towards understanding enigmatic fungal-viral interactions and the different abundances of metabolic proteins are interesting. Overall the authors make sound interpretations of the results across various experiments, and it is a worthy study. My primary suggestion is that the authors revisit the figures in order to make them publication-ready and restructure the figures to ensure the most important ones are primary and additional are supplemental. An additional map figure would add interesting context. Specific comments to follow:

Major:

Figure 1 should be supplemental.

Figure 2 It may be more useful to the reader to define the pfams. For instance, you could add a legend to define the pfams (05919 = mitovir pol, etc.) I'm sure there are additional ways to make this information more reader-available, too.

Figure 8 A- valuable information. B-What is this showing? Why is this important/what value do you see this adding to the paper? C- I don't understand the comparison..... it looks like it is comparing the area of isolates grown at 5dpi to isolates grown at 3dpi... which doesn't make any sense.

Overall (fig 8):Can you find an alternative way to present these data? Maybe delta, or growth rate? The statistics reported are not able to be visualized by these figures (lines 589, 592-594, etc). Be sure to label axes in the new figure.

Figure 9 A & C- Would be more clear to use the traditional circling approach to demonstrate significant differences between groups. Would be more visually clear and appealing to have a legend with the isolates and their corresponding colors than have the text just near the data points as is.

Minor:

Line 21 strike extra use of word "conifer"

Lines 56-60 This left me feeling some conflict. *Heterobasidion* is a problem... but not a very big problem relevant to bark beetle? Would it be accurate to articulate that the challenge lies in how bark beetle overshadows *heterobasidion* in terms of damage? Or, is bark beetle SPREADING *heterobasidion*? Please clarify.

Line 504-505 Unclear what the "two bigger clusters" that are "grouped together" is referring to.

Line 507 I think this is just meaning because of their existence in hosts with additional infections there is now co-infection, but it reads as somehow the ambi viruses *facilitate* co-infection with multiple viruses. Please adjust for clarity.

General Have you consider any sort of spatial analyses? Even a map of sampling sites and viruses at each would be interesting; this map could also include results from other studies. Not required, but would add valuable context.

General Comments:

The manuscript titled "Diversity and impact of single-stranded RNA viruses in Czech *Heterobasidion* populations" presents an interesting study on the virome of *Heterobasidion* populations. While the study offers valuable insights, certain aspects require clarification to enhance the overall readability and reproducibility of the findings.

Major Concerns:

The RNA-seq analysis methodology described in the manuscript is somewhat unclear. It is essential to provide a detailed step-by-step description of the RNA-seq workflow, including the following:

-The specific criteria used for quality control and filtering of raw reads.

“The quality of raw reads was assessed using FastQC-0.11.8 (<https://www.bioinformatics.babraham.ac.uk/projects/fastqc>, accessed on 11 April 2022). Adapter sequences were clipped with cutadapt-3.4 (Martin, 2011) and zero-length reads discarded.”

- Detailed information about the bioinformatic pipelines used for virus discovery, including versions of software and specific parameters/settings.

“The first bioinformatic pipeline consisted of *de novo* assembly with Trinity-2.11.0 (Grabherr et al., 2011) using RF for --SS_lib_type flag and searching for similarities of the obtained contigs to custom virus and host protein and nucleotide reference databases obtained from NCBI, using the BLASTX and BLASTN algorithms of the BLAST+-2.10.0 program (Camacho et al., 2009).”

- The criteria for selecting viral contigs and the subsequent steps for validation.

The following sentence was added: Final viral contigs were selected based on sequence similarity to known viruses, identification of conserved viral marker genes (e.g., capsid protein, RdRp) and coverage depth across the contig to ensure it represents a complete or near-complete viral genome.

-An explicit explanation of how the pooling of RNA samples was handled and how it may affect the results.

“Three pools consisting of total RNA of 16 *H. annosum* strains, another 16 *H. annosum* strains and 13 *H. parviporum* strains (Table S1) were prepared as follows. Equal volumes of each RNA sample were mixed, diluted to 200 ng/μL in 70 μL volume, treated with the TURBO DNA-free™ Kit (Thermo Fisher Scientific) and sent to SEQme s.r.o. (Dobříš, Czechia).”

The methodology section needs revision to include these details to ensure other researchers can replicate the study.

-The manuscript lacks a detailed description of the models used in the phylogenetic analysis. The selection of appropriate models is crucial for the reliability of phylogenetic inferences.

I recommend identifying the models used for phylogenetic analysis and providing a rationale for their selection. Additionally, it includes the software and versions used for constructing phylogenetic trees.

The model used is described in the figure captions as it differed for each individual tree. The software and version (MEGA11) is indicated in the text as well as in the figure captions.

“The model with the lowest Bayesian Information Criterion score was selected for each tree.”

-The figure legends provided are insufficient for readers to understand the presented data fully. Each figure should be self-explanatory, providing enough context and detail about what is being shown.

The figure legends need to be elaborated to include explanations of symbols, scales, and annotations. Ensure that each figure can stand alone without requiring the reader to refer to the main text.

We have revised every figure caption and tried to provide clear explanations.

Specific Comments:

In Section 2.4 - Bioinformatics:

-It would be helpful to clarify the two different approaches combined for in silico data mining. Specify if these approaches were applied to all datasets or if there were variations.

“Two approaches were combined for *in silico* data mining for viruses in all of the NGS datasets, each of which consisted of ca. 600 million reads.”

In Table 1 - Virus Detection:

-Include additional columns in the table to specify the coverage and depth of each detected virus. This will help readers assess the reliability of the detection.

We have revised Table 1 to ensure the mapping reads and the mean depth are included.

In Supplemental Figures:

-Ensure that all supplemental figures are referenced appropriately in the main text. It is currently unclear how some supplemental data supports the presented findings.

We have checked every reference to supplemental figures throughout the manuscript to make sure all of them are correctly referenced, and also added one paragraph to the end of the manuscript listing all supplemental data.

Reviewer #2 (Comments for the Author):

This study documents the mycoviruses inhabiting an important and impactful fungal pathogen of conifers in Czechia. The authors contextualize their mycoviral findings with expectations given the Heterobasidion viruses in other parts of Europe, and interestingly found very little similarity in viromes. The most unique, and perhaps novel, aspect of the study was the proteomics work, which is rarely done in mycovirus research but is an important future direction of research. While the results of the proteomic profiling may not have been what was expected since no viral proteins were detected, it is an important step towards understanding enigmatic fungal-viral interactions and the different abundances of metabolic proteins are interesting. Overall the authors make sound interpretations of the results across various experiments, and it is a worthy study. My primary suggestion is that the authors revisit the figures in order to make them publication-ready and restructure the figures to ensure the most important ones are primary and additional are supplemental. An additional map figure would add interesting context. Specific comments to follow:

Major:

Figure 1 should be supplemental.

Fixed.

Figure 2 It may be more useful to the reader to define the pfams. For instance, you could add a legend to define the pfams (05919 = mitovir pol, etc.) I'm sure there are additional ways to make this information more reader-available, too.

Fixed.

Figure 8 A- valuable information. B-What is this showing? Why is this important/what value do you see this adding to the paper? C- I don't understand the comparison..... it looks like it is comparing the area of isolates grown at 5dpi to isolates grown at 3dpi... which doesn't make any sense.

B shows the absolute growth of isolates on malt extract agar plates in mm² (mean and range of values of five technical replicates) at two points in time, used as the metric to compare the fitness of *a priori* determined pairs of isolates. C- The diagonal black "line" divides the square to two triangles, which should be viewed separately. The upper right triangle includes comparisons 3 dpi and the bottom left triangle pertains to 5 dpi. The caption has been modified to facilitate understanding.

Overall (fig 8): Can you find an alternative way to present these data? Maybe delta, or growth rate? The statistics reported are not able to be visualized by these figures (lines 589, 592-594, etc). Be sure to label axes in the new figure.

For wood-decay fungi, the rapid colonization of substrates is a crucial trait while competing with other fungi in nature. Fast mycelial growth enables the capture of abundant nutrition sources and ensures survival in the early phase of life when a large mycelium is not yet formed. Therefore, we believe that absolute growth is the relevant metric for this study, which is consistently used in our literature, thus enabling comparison of results across different studies. The statistical significance of differences in the growth rate is visualized by Fig. 7C,

also showing which pairs of isolates were compared to each other and which were not tested. Axes have been labelled.

Figure 9 A & C- Would be more clear to use the traditional circling approach to demonstrate significant differences between groups. Would be more visually clear and appealing to have a legend with the isolates and their corresponding colors than have the text just near the data points as is.

Fixed.

Minor:

Line 21 strike extra use of word "conifer"

“*Heterobasidion annosum* sensu lato comprises some of the most devastating pathogens of conifers.”

Lines 56-60 This left me feeling some conflict. Heterobasidion is a problem... but not a very big problem relevant to bark beetle? Would it be accurate to articulate that the challenge lies in how bark beetle overshadows heterobasidion in terms of damage? Or, is bark beetle SPREADING heterobasidion? Please clarify.

“Heterobasidion root rot has been a serious problem in planted Norway spruce (*Picea abies*) and Scots pine (*Pinus sylvestris*) stands in Czechia ever since forest management was introduced (Černý, 1989; Sedlák and Tomšovský, 2014), and is one of the significant factors that exposes trees to European spruce bark beetle (*Ips typographus*) infestation (Wahlman, 2024). In 2020, the volume of salvage fellings in Czechia rose to an unprecedented 34 million m³, mostly due to *I. typographus* as the main disease agent (Ministry of Agriculture of the Czech Republic, 2021).”

Line 504-505 Unclear what the "two bigger clusters" that are "grouped together" is referring to.

“The ambi-like viruses of *Heterobasidion* are presumably of polyphyletic origin as they were placed in three separate branches (Fig. 5). The majority belong to the *Trimbiviridae* family, a smaller group to *Dumbiviridae*, whereas HetAlV6 fell into *Quambiviridae*, closely associated with *Armillaria novae-zelandiae* ambi-like virus 1 (Linnakoski et al., 2021).”

Line 507 I think this is just meaning because of their existence in hosts with additional infections there is now co-infection, but it reads as somehow the ambi viruses facilitate co-infection with multiple viruses. Please adjust for clarity.

“With two exceptions (isolates 1991 and 2040), the *Heterobasidion* ambi-like viruses were in coinfections in their host strains with CVMPIS, HetFV1, ambi-, narna- or ourmia-like viruses (Table 1).”

General

Have you consider any sort of spatial analyses? Even a map of sampling sites and viruses at each would be interesting; this map could also include results from other studies. Not required, but would add valuable context.

We appreciated and considered the suggestion. As part of a purely exploratory study, the goal of the sampling was simply to obtain a decent number of isolates for screening in a short amount of time. Thus, the sampling was not systematic; it was arbitrary with logistics being the primary factor taken into account. After detecting viruses, we were wondering if there could be some spatial pattern in their distribution. Mapping the isolates did not reveal any pattern. Unfortunately, our data is too limited to perform a meaningful population genetic analysis (only 45 strains have been screened for viruses' presence and just 12 of these were infected). Moreover, the South Moravian Region was heavily overrepresented among the sampling localities. Due to the unequal distribution of samples, a map would be overcrowded at spots, visually unappealing and counterproductive considering it would not illustrate any interesting point.

REFERENCES

- Camacho, C., Coulouris, G., Avagyan, V., Ma, N., Papadopoulos, J., Bealer, K., Madden, T.L., 2009. BLAST+: architecture and applications. *BMC Bioinformatics* 10, 421. <https://doi.org/10.1186/1471-2105-10-421>
- Černý, A., 1989. Parazitické dřevokazné houby [Parasitic wood-decaying fungi]. Státní zemědělské nakladatelství, Prague, Czechia.
- Grabherr, M.G., Haas, B.J., Yassour, M., Levin, J.Z., Thompson, D.A., Amit, I., Adiconis, X., Fan, L., Raychowdhury, R., Zeng, Q., Chen, Z., Mauceli, E., Hacohen, N., Gnirke, A., Rhind, N., di Palma, F., Birren, B.W., Nusbaum, C., Lindblad-Toh, K., Friedman, N., Regev, A., 2011. Full-length transcriptome assembly from RNA-Seq data without a reference genome. *Nat. Biotechnol.* 29, 644–652. <https://doi.org/10.1038/nbt.1883>
- Linnakoski, R., Sutela, S., Coetzee, M.P.A., Duong, T.A., Pavlov, I.N., Litovka, Y.A., Hantula, J., Wingfield, B.D., Vainio, E.J., 2021. *Armillaria* root rot fungi host single-stranded RNA viruses. *Sci. Rep.* 11, 7336. <https://doi.org/10.1038/s41598-021-86343-7>
- Martin, M., 2011. Cutadapt removes adapter sequences from high-throughput sequencing reads. *EMBnet.journal* 17 (1), 10–12. <https://doi.org/10.14806/ej.17.1.200>
- Ministry of Agriculture of the Czech Republic, 2021. Zpráva o stavu lesa a lesního hospodářství České republiky v roce 2020 [Report of the state of forests and forestry in Czechia 2020]. Prague, Czechia.
- Sedlák, P., Tomšovský, M., 2014. Species distribution, host affinity and genetic variability of *Heterobasidion annosum sensu lato* in the Czech Republic. *For. Pathol.* 44 (4), 310–319. <https://doi.org/10.1111/efp.12102>
- Wahlman, W., 2024. The effect of *Heterobasidion* root rot on *Ips typographus* infestation risk on Norway spruce. University of Helsinki, Finland.

Re: mSystems00506-24R1 (Diversity and impact of single-stranded RNA viruses in Czech Heterobasidion populations)

Dear Dr. Leticia BOTELLA:

Your manuscript has been accepted, and I am forwarding it to the ASM production staff for publication. Your paper will first be checked to make sure all elements meet the technical requirements. ASM staff will contact you if anything needs to be revised before copyediting and production can begin. Otherwise, you will be notified when your proofs are ready to be viewed.

Sincerely,
Liyuan Ma
Editor
mSystems

Reviewer #2 (Comments for the Author):

The authors have adequately addressed my concerns.